# Effect of radiation interaction and aerosol processes on ventilation and aerosol concentrations in a real urban neighbourhood in Helsinki

Jani Strömberg[1], Xiaoyu Li[1], Mona Kurppa[2], Heino Kuuluvainen[3], Liisa Pirjola[1,4], and Leena Järvi[1,5]

[1]Institute for Atmospheric and Earth System Research, Faculty of Science, University of Helsinki, Helsinki, Finland
[2]Kjeller Vindteknikk, Espoo, Finland
[3]Aerosol Physics Laboratory, Physics Unit, Faculty of Engineering and Natural Sciences, Tampere University, Tampere, Finland
[4]Department of Automotive and Mechanical Engineering, Metropolia University of Applied Sciences, Vantaa, Finland
[5]Helsinki Institute of Sustainability Science, Faculty of Science, University of Helsinki, Helsinki, Finland

**Correspondence:** JANI STRÖMBERG (jani.stromberg@helsinki.fi)

**Abstract.** Large eddy simulation (LES) is an optimal tool to examine aerosol particle concentrations in detail within urban neighborhoods. The concentrations are a complex result of local emissions, meteorology, aerosol processes, and local mixing conditions due to thermal and mechanical effects. Despite this, most studies have focused on simplification of the affecting processes such as examining the impact of local mixing in idealised street canyons or treating aerosols as passive scalars. The aim of this study is to include all these processes into LES using PALM, and to examine the importance of radiative heating and aerosol processes in simulating local aerosol particle concentrations and different aerosol metrics within a realistic urban neighborhood in Helsinki under morning rush hour with calm wind conditions. The model outputs are evaluated against mobile laboratory measurements of air temperature and total particle number concentration ($N_{tot}$), and drone measurements of lung deposited surface area (LDSA).

The inclusion of radiation interaction in LES has a significant impact on simulated near surface temperatures in our study domain increasing them on average from 8.6 °C to 12.4 °C. The resulting enhanced ventilation reduces the pedestrian level (4 m) $N_{tot}$ by 53%. The reduction of $N_{tot}$ due to aerosol processes is smaller, only 18%. Aerosol processes impact particularly the smallest particle range, whereas radiation interaction is more important in the larger particle range. The inclusion of radiation interaction reduces the bias between the modelled and mobile laboratory measured air temperatures from -3.9 °C to +0.2 °C, and $N_{tot}$ from +98% to -13%. With both aerosol and radiation interaction on, the underestimation is 16 %, which might be due to overestimation of the ventilation. The results show how inclusion of radiative interaction is particularly important in simulating PM$_{2.5}$ whereas aerosol processes are more important in simulating LDSA in this calm wind situation.

## 1 Introduction

Urban air pollution has been recognized to be one of the major global challenges as it has been estimated to result annually up to 0.8 million premature deaths in Europe (Lelieveld et al., 2019) and 3 million deaths worldwide (Lelieveld et al., 2015; WHO,

2016). The numbers are expected to increase further in future as the proportion of global population living in urban areas is projected to increase from current 55% (2018) to 68% by 2050 (United Nations, 2019). Often the poorest air quality is observed at pedestrian level in street-canyons due to vicinity of road traffic and degraded ventilation (Kurppa et al., 2020). Ventilation of a street-canyon or a wider urban area depends on building morphology but also on radiative processes resulting in increased turbulent production and mixing of air when solar radiation is present (Tominaga and Stathopoulos, 2013; Nazarian and Kleissl, 2016; Park et al., 2017). Mechanically and thermally driven turbulence affect urban air pollutant concentrations and thus understanding their behaviour in detail are vital for correct estimation of urban air quality and aerosol particle concentrations.

Turbulence and street canyon flows have been researched intensively in recent years through computational fluid dynamics (CFD) modelling (e.g. Letzel et al., 2012; Park and Baik, 2013; Kwak et al., 2015; Kurppa et al., 2020). From the two main modelling methods, Reynolds-averaged Navier–Stokes (RANS) and Large-Eddy Simulation (LES), LES has been found to perform better in resolving instantaneous turbulence in realistic complex urban settings (Salim et al., 2011; García-Sánchez et al., 2018). In LES the subgrid scale turbulence is parameterised but otherwise the three-dimensional wind field and scalar variables describing boundary layer flows are solved with high spatial and temporal resolution (Maronga et al., 2020). LES has been used to examine the impact of thermal effects on urban ventilation but mainly in idealised street canyons. Nazarian and Kleissl (2016) examined an idealised 3x3 building array, where the differential heating was parameterised with a new dimensionless universal Richardson number, and found that the roof and ground heating alter the formed canyon vortices' location and strength. Nazarian et al. (2018) had a similar simulation setup and results about the changes to street vortices' structures Nazarian and Kleissl (2016), but additionally concluded that the spatial pollutant field is not as strongly affected by the heating distribution as the vertical removal of pollutants from the street canyons. Most simulations in realistic urban settings have not yet included the effect of solar radiation induced thermal turbulence on street canyon flow patterns and ventilation, and furthemore on pollutant concentrations mainly due to the computational cost of LES (e.g. Salim et al., 2011; Kurppa et al., 2020; Karttunen et al., 2020). Park and Baik (2013) had simplified radiation scheme to study turbulent coherent structures between no-heating and bottom heating cases . Implementing a radiation scheme to an LES model offers a way to model the complex radiative transfer processes in urban areas in detail, such as multiple reflections, diffuse radiation and the effect of shading, and further the resulting thermal effects on flow structures (Resler et al., 2017).

Furthermore, for realistic simulation of air pollutants and particularly aerosol particle concentrations, aerosol particle dynamics accounting for their chemical and physical processes need to be considered (Kurppa et al., 2019, 2020). Traditionally, pollutants and aerosol particles have been treated as passive scalars in CFD (Branford et al., 2011; Gousseau et al., 2011; Cai, 2012; Tominaga and Stathopoulos, 2013) and only a few LES models allow detailed description of aerosol particles, their size distributions and dynamic processes. Steffens et al. (2013) included a CFD-based Comprehensive Turbulent Aerosol Dynamics and Gas Chemistry model (CTAG) in their simulations. Kurppa et al. (2019) implemented Sectional AerosoL Module SALSA into PALM. Zhong et al. (2020) on the other hand used WRF-LES to model the behavior of UFPs on a neighbourhood scale.

The main aims of this study are to conduct a novel LES including both aerosol processes and mixing conditions within a real urban neighborhood and to examine the impact of radiative effects and aerosol dynamics on different aerosol particle metrics in an built-up neighbourhood in Helsinki under calm wind conditions during morning rush hour. The relative importance

of including radiative effects and aerosol processes in simulating aerosol particle concentrations and distributions describing different aerosol metrics will be examined. This is made as simplification in LES can save computational resources and for this it is important to understand the relative importance of different processes. The LES model PALM (Maronga et al., 2020) will be used in the simulations as it allows for realistic description of the urban surface and aerosol dynamics, and it can be coupled with the radiation scheme RRTMG (Rapid Radiative Transfer Model for Global models, Krč et al., 2021), which uses RTM (Radiative Transfer Model) to model the radiation interaction within the urban canopy layer. Helsinki was chosen due to the intensive observational air quality campaign made within the study area allowing extensive model evaluation. Model outputs will be compared against LDSA of aerosol particles measured using a drone, and total particle number concentration and air temperature measured using a mobile laboratory (Järvi et al., 2023).

## 2  Methods

### 2.1  PALM

PALM is an LES model used to study atmospheric and oceanic boundary layer dynamics (Maronga et al., 2020). PALM solves the non-hydrostatic, filtered and incompressible Navier–Stokes equations of wind ($u$, $v$, and $w$) and scalar variables, including turbulent kinetic energy, potential temperature and specific humidity, in a Boussinesq-approximated form. As the model is well-scalable on massively parallel computer architectures, it is particularly well-suited for urban simulations on domains up to a city-scale with a fine grid resolution. PALM has also a self-nesting capability allowing a fine grid resolution within the main domain of interest (i.e. child domain), and coarser resolution in parent and root domains allowing large total modelling domain (Hellsten et al., 2021). Moreover, PALM has multiple features which enhance its usability to examine urban turbulence. First of all, it utilises surface models such as the Land-Surface Model (LSM) and the Urban Surface Model (USM) solving the energy balance for each surface (Resler et al., 2017; Gehrke et al., 2020). LSM requires the use of a radiation scheme, which is provided by the external RRTMG library embedded in PALM and enabled in these simulations (Krč et al., 2021). Secondly, PALM has a plant canopy model (PCM) which is used to model the interaction between vegetation and flow (Karttunen et al., 2020). Finally, the Sectional Aerosol module for Large Scale Systems (SALSA) is used to solve the aerosol processes responsible for modifying the size distribution and pollutant interaction with the surface in PALM (Kurppa et al., 2019, 2020). The most important modules for this study are RRTMG and SALSA.

### 2.1.1  RRTMG

RRTMG (Rapid Radiative Transfer Model for Global models) is an external library, which can be used with PALM to provide the variables responsible for describing the radiative interaction between the surface and the atmosphere. RRTMG is used as a single-column model in PALM, whereas a separate multi-reflection RTM (Radiation Transfer Model) is used within the urban canopy layer (Resler et al., 2017). RRTMG feeds the RTM, which is used by the surface models USM and LSM, with the necessary components such as the time of day and coordinates to solve the energy balance over all surfaces (Resler et al., 2017;

Salim et al., 2022; Gehrke et al., 2020). RTM is capable of calculating multiple reflections, diffuse radiation and absorbed radiation on different surfaces (Krč et al., 2021). RTM handles plant canopies as fully transparent in the longwave radiation range and therefore shading is only modelled for the shortwave range in these cases (Resler et al., 2017). Sky-view factors are calculated at each radiation timestep and on both vertical and horizontal grid points, which describe the amount of sky visible from a given surface as a fractional number between 0 - 1 (Salim et al., 2022; Krč et al., 2021).

### 2.1.2 SALSA

The sectional aerosol module SALSA (Kokkola et al., 2008), which has embedded in the PALM model system (Kurppa et al., 2019), is employed to describe the aerosol population by discretizing the aerosol number size distribution into several size bins based on the geometric mean dry particle diameter. Each bin can be composed of different chemical components including sulphate, organic carbon, black carbon, nitrate, ammonium, sea salt, mineral dust, and water. The hybrid bin method is used for the update of aerosol size distribution in both two subranges (Kokkola et al., 2018). SALSA is designed to resolve aerosol microphysical processes in a very large number of grid points, comprising of nucleation, coagulation, condensation, dissolution, as well as dry deposition on horizontal and vertical surfaces and resolved-scale vegetation. The implementation of SALSA is flexible so that the user can decide the number of size bins, diameter range of aerosols, specific chemical components, and involved aerosol dynamic processes.

### 2.2 Model setup

Our study area is a 42-m wide street canyon (average height to width H/W=0.45) and its immediate surroundings in Helsinki, Finland (Figure 1), on an early summer morning on the 9th of June, 2017. The street canyon has pavement and three lanes for both directions with the outermost lanes next to the pavement reserved for public transport. In the middle there are two tram lines with street tree rows separating them from the lanes. An urban air quality monitoring supersite operated by the Helsinki Region Environmental Services Authority is located on the southern side of the street canyon. The simulation setup consists of a root (768,768 cells), parent (768,768 cells) and child (576, 576 cells) domain, each with an increase in horizontal resolution by a factor of 3 when moving from root (9 m) to the smaller domains (3 m in parent and 1 m in child). The surface energy balance and flow are solved in each domain whereas SALSA is only enabled in the child domain. Dynamic boundary conditions are supplied by numerical weather prediction data from the MetCoOp Ensemble Prediction System (MEPS, Bengtsson et al., 2017; Müller et al., 2017) which provides the necessary forcing for initializing the large scale motions in the atmosphere. The trajectory model for Aerosol Dynamics, gas and particle phase CHEMistry (ADCHEM, Roldin et al., 2011b), provides the background trace gas concentrations, particle number concentrations and chemical composition of aerosol particles for SALSA. Emissions from road traffic within the child domain are estimated by combining information on hourly vehicle fleet composition, and particle number and gaseous unit emissions factors. In this study, we only considered exhausted traffic emissions and did not involve emissions from vegetation. In SALSA the aerosol particle size distribution is described by 10 bins ranging from 2.5 nm to 1 $\mu$m, and the particles contain sulphate, organic carbon, black carbon, nitrate, and ammonium. The

aerosol processes of condensation, coagulation, and dry deposition are included and calculated per second. More information
on the model setup can be found from Kurppa et al. (2020).

RRTMG uses information about the usage and built time-period of buildings (Appendix A1), pavement materials (Appendix A3) and vegetation types (Appendix A2) to model the thermal and radiative properties of each surface. Building usage for our study area and the time-period they were built were obtained from the City of Helsinki open database (HRI, 2017, see Supplementary material). Soil information (Appendix A4) needed by the LSM was retrieved from the national land survey of Finland (GTK, 2018).

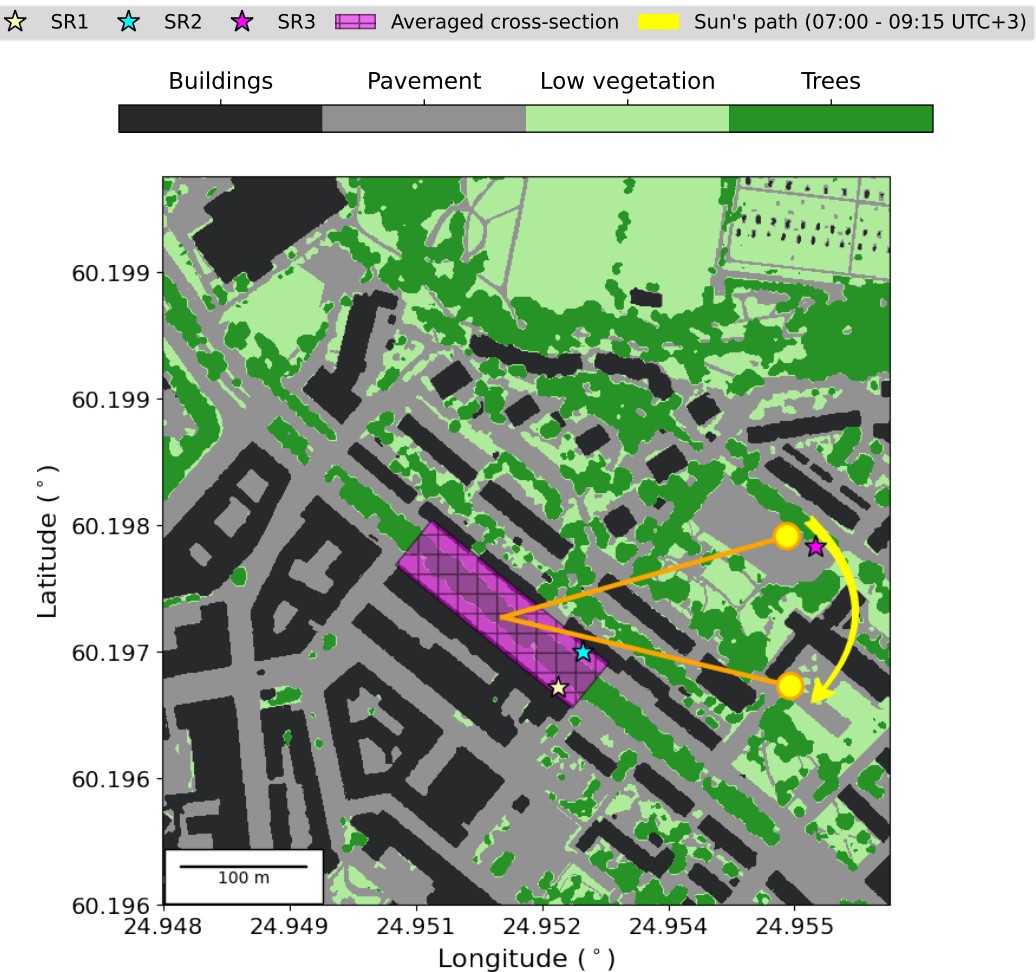

**Figure 1.** Child domain of the model setup including the location of three statistical regions (SRs), area used to calculate street canyon cross-section and the Sun's path during the main run.

## 2.3 Model runs

In order to understand the effect of radiation (R) and aerosol processes (A) on aerosol particle distributions both together and separately, four model runs are performed. $R_0A_0$ is the base run, with both aerosol processes and radiation turned off. This run purely solves the flow and aerosol dispersion as passive scalars, and the spatial distribution of particles and ventilation is only
affected by the mechanical processes. $R_0A_1$ increases the level of complexity by including the aerosol processes (condensation, coagulation and deposition) while leaving the radiation out. $R_1A_0$ on the other hand has radiation turned on, but only simulates the transport of the aerosols, without any aerosol processes affecting their size distribution. $R_1A_1$ combines the two processes together and simulates both radiation and aerosol processes being the most complex in terms of the amount of processes affecting the simulation. $R_0A_1$ is the same model setup as in Kurppa et al. (2020). In terms of atmospheric stability, $R_0A_0$ and
$R_0A_1$ describe purely neutral cases, whereas $R_1A_0$ and $R_1A_1$ are unstable. The neutral simulations do not have USM and LSM since they require a radiation scheme. This means that the temperatures are directly provided by MEPS dynamic input in the neutral cases. The PALM revision used is r4734.

**Table 1.** Summary of the model runs discussed in this study. R (radiation) and A (aerosol processes) describe the changing conditions between the runs and the subscript under them tells if that part of the simulation is turned on (1) or off (0).

| Model runs | $R_0A_0$ | $R_0A_1$ | $R_1A_0$ | $R_1A_1$ |
|---|---|---|---|---|
| Radiation on | ✗ | ✗ | ✓ | ✓ |
| Aerosol processes on | ✗ | ✓ | ✗ | ✓ |

## 2.4 Initialisation

Simulation time of the model runs is separated into three parts: spin-up, precursor and main run (Figure 2). The main run of
all modelled processes including SALSA covers the time period 07:00 - 09:15 UTC+3, which was chosen as observations from an intensive observational campaign of local air quality are available for model evaluation (Järvi et al., 2023). Before the main run, a precursor run to initialise flow and turbulence (06:00 - 07:00 UTC+3) is performed (Kurppa et al., 2020). From the restart data provided by the precursor run, PALM is able to start the main run to get the final output data. In addition, the runs with radiation enabled require an additional spin-up run of 24 hours (full diurnal cycle of solar radiation) for realistic
development of surface temperatures to accurately model the heat exchanges with the atmosphere (Resler et al., 2017; Krč et al., 2021). For the spin-up run (8 June 06:00 – 9th June 06:00 UTC+3), PALM needs the mean and variation amplitude of the potential temperature, which were calculated to be 12°C and 3°C using the Finnish Meteorological Institute (FMI, 2017) weather station data, and taking the 24-hour mean temperature and difference between minimum and maximum temperatures during the precursor period. The precursor run period is cloudy, with little diurnal variation in air temperature (Figure 2). The
morning at the time of the main run is less cloudy and solar radiation is already at the same level at 8 am as the maximum of the

previous day. Wind is from the west and during the main run, the wind turns from 273° to 305° and the wind speed increases due to the rising sun.

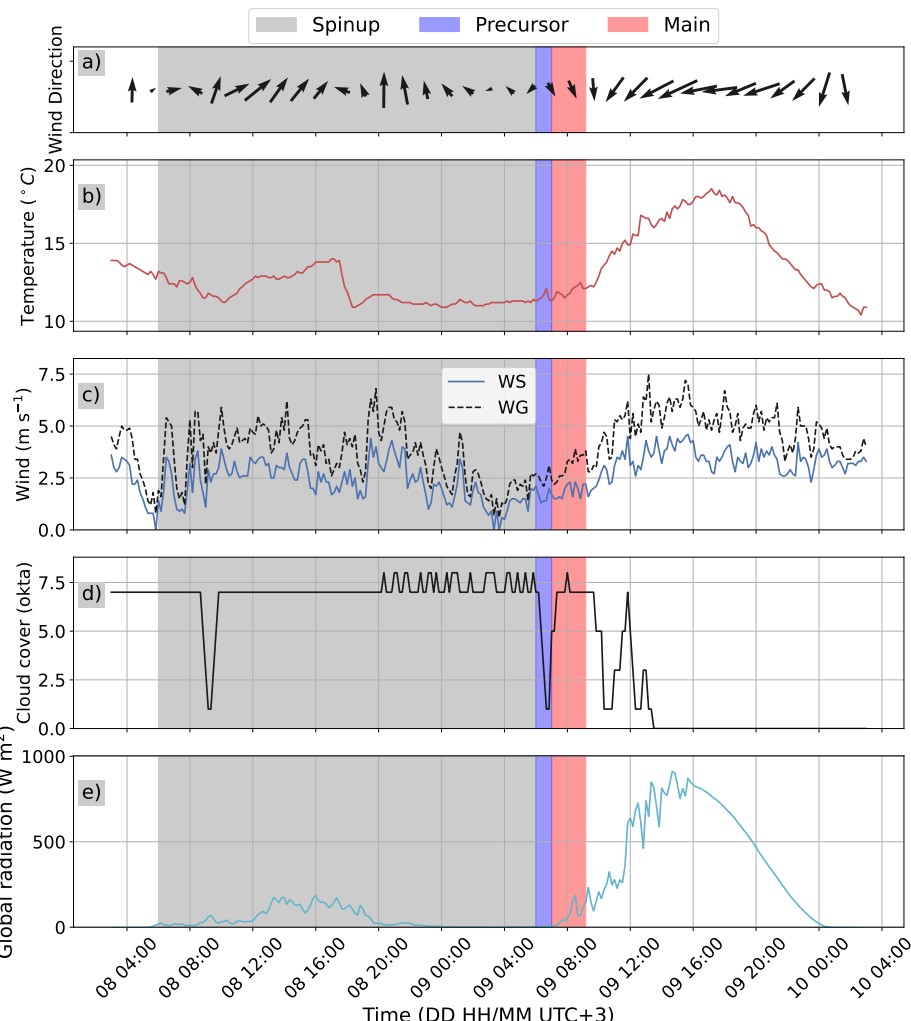

**Figure 2.** Meteorological conditions during the whole simulation period as measured on the Finnish Meteorological Institute weather station located at Kumpula every 10 minutes. Variables plotted are wind arrows (a), temperature ($T$) (b), wind speed ($WS$) and gusts ($WG$) (c), cloud cover (d) and global radiation (e). Grey area highlights the spinup run, blue the precursor run and red the main run.

## 2.5 Data analysis

Data with greater temporal and spatial resolution than what is saved from the full domains are saved from specific statistical
regions between 0.1 s intervals at 1 m resolution. In our analysis we use four such regions (see also Figure 1). Statistical
regions SR1 and SR2 cover 5 m×5 m areas from ground level to height of 144 m on opposite sides of the main street canyon
named hereafter supersite and opposite supersite (Kurppa et al., 2020). Similarly, SR3 is a 5 m×5 m×144 m column repre-
senting a background measurement site away from the main street canyon. All three statistical regions save profile information
about wind components, fluxes, air temperature and available SALSA output of particle number size distribution, total particle
number concentration ($N_{tot}$), number concentration for ultrafine particle (UFP, particles with aerodynamic diameter less than
0.1 $\mu$m), particulate mass for particles with aerodynamic diameter below 2.5 $\mu$m (PM$_{2.5}$), and LDSA. Different aerosol metrics
are analysed as they reflect the characteristics of particles with different sizes. It has been reported that dispersion is somewhat
different for smaller and larger particles (Rivas et al., 2017; Karttunen et al., 2020). The statistical regions SR1 and SR2 are
chosen to get model output data with higher temporal resolution enabling comparison to LDSA observations made using a
drone, and SR3 to get comparison how the vertical profiles of aerosol particles look like outside the emission sources.

Additionally, a fourth area of interest is calculated for the cross-section of the main street canyon covering a 176 m long
section (Figure 1). This is not saved as a separate statistical region but rather taken as a subset from a mask area slightly smaller
than the child domain containing only the main street and its immediate surroundings. The street canyon section is chosen as
it follows the long building on the southwestern side and is between two side streets coming from the west. The area is chosen
to provide an overall understanding of the flow and aerosol fields within the street canyon. The mean cross-section for this
area is calculated by applying a 51° coordinate rotation to the horizontal wind components in order to align the street with the
y-coordinate direction. The cross-sectional wind analysis and $N_{tot}$ analysis use data from 2 to 32 meters above the ground that
follow the terrain. Analysed data cover the time period 7:00-9:15 when only modelled data are analysed, and time period 7:15-
9:15 when modelled data are compared to drone observations. Additionally, colourblind-friendly colour maps were provided
by Crameri (2021).

## 2.6 Observations

During the measurement campaign, observations with a mobile laboratory Sniffer (Pirjola et al., 2004) and a drone were
conducted (Järvi et al., 2023). The mobile laboratory measured $N_{tot}$ of particles with 2.5–20 nm in aerodynamic diameter
using a Condensation Particle Counter (CPC, TSI 3776, TSI Ltd, USA) and a 2-m air temperature using a temperature and
humidity probe (HMP45A, Vaisala Oyj, Finland) with a 1-s resolution. The latter sensor measures temperature at range -40–
60°C with accuracy of 0.2°C (in 20°C) and relative humidity at range 0-100% with accuracy of ±0.1% RH. The inlet for
the aerosol instrument was located above the windshield at 2.4 m. The van speed and position were recorded using a global
positioning system (model GPS V, Garmin). The van was driving along the main street canyon and its side streets, and standing
at the background, supersite and opposite supersite (i.e. matching with the statistical regions). In the standing measurements,
the first 3-min were always excluded from the data analyses in order to avoid contamination coming from the van itself. Mean

temperature and $N_{\text{tot}}$ were calculated for 5 m x 5 m grids for 07:15–09:15 within the area where the mobile laboratory was driving (Kurppa et al., 2020). The grid size was determined based on the particle number concentration measurements and number of counts in each grid and by the width of streets.

At the same time with the mobile laboratory, a multi-rotor drone (X8, VideoDrone Finland Ltd) was measuring the vertical distribution of the alveolar LDSA of aerosol particles using an electrical particle sensor (Partector, Naneos GmbH, Switzerland). The measurements were done within the statistical subregions SR1 and SR2 located on both sides of the street canyon. The drone flew 10 times up and down between $z = 2$ and 50 m during one 30-min measurement interval, after which measurements were repeated on the other side. Geometric mean profiles from the 10 repetitions for the supersite (opposite supersite) were calculated for time periods 07:16-07:44 (07:54-08:14) and 08:23-08:44 (08:51-09:15). More details of the drone measurements and data analysis can be found from Kuuluvainen et al. (2018).

## 3 Results and discussion

### 3.1 Near surface air temperature

Figure 3 shows the overall change in the mean near surface air temperature at a height of 2 metres ($T_{2m}$) for the base run $R_0A_0$ without radiation interaction and aerosol processes, and for the model run with both aerosol and radiation interaction on ($R_1A_1$) within the child domain. In the base run $R_0A_0$, the mean $T_{2m}$ over the child domain is 8.6°C and varies spatially between 8.4 to 8.9°C (Figure 3a) whereas in $R_1A_1$, the mean overall $T_{2m}$ is 12.4°C, and ranges spatially between 11.2 and 17.9°C. Thus, there is an average increase of 3.8°C within the entire child domain in $R_1A_1$ when compared to $R_0A_0$. The largest $T_{2m}$ increase is observed close to the eastward facing building walls in the main street canyon. This is due to the early morning sun heating the walls and creating a more heterogeneous temperature distribution compared to $R_0A_0$. Aerosol processes do not affect radiation, but they impact the flow (Sühring, 2022), which in turn affects near surface air temperatures. This impact is however minor with -0.6% difference in $T_{2m}$ between $R_1A_0$ and $R_1A_1$. Thus, we can say that the difference between $R_0A_0$ and $R_1A_1$ is caused by radiation interaction.

The observed $T_{2m}$ measured by the mobile laboratory are in the range of 12-13°C with a spatial mean of 12.4°C (Figure 4a). When compared to the modelled temperatures (Figure 4b,c), $R_0A_0$ underestimates $T_{2m}$ by 3.9°C due to lack of heat exchange between the surface and atmosphere, and the absence of solar radiation (Gehrke et al., 2020). When comparing these to $R_1A_1$, an immediate improvement is visible, with a decrease in the bias from -3.9°C to +0.2°C. With radiation interaction turned on, the overall spatial distribution of air temperature falls in the same range at 12.7°C for the whole area. This shows the model describes heating of the surfaces correctly. There are some areas with slightly larger air temperatures compared to the observations close to western side wall next to the supersite. Similar behaviour of the highest temperatures on one side of a street canyon is also reported in Jiang and Yoshie (2018). This can be attributed to the amount of solar radiation received, as a large gap between buildings on the opposite side of the street combined with low azimuth angles of the sun cause this strip of wall to receive more incoming shortwave radiation than the rest of Mäkelänkatu (Oke, 1988). The used revision of PALM

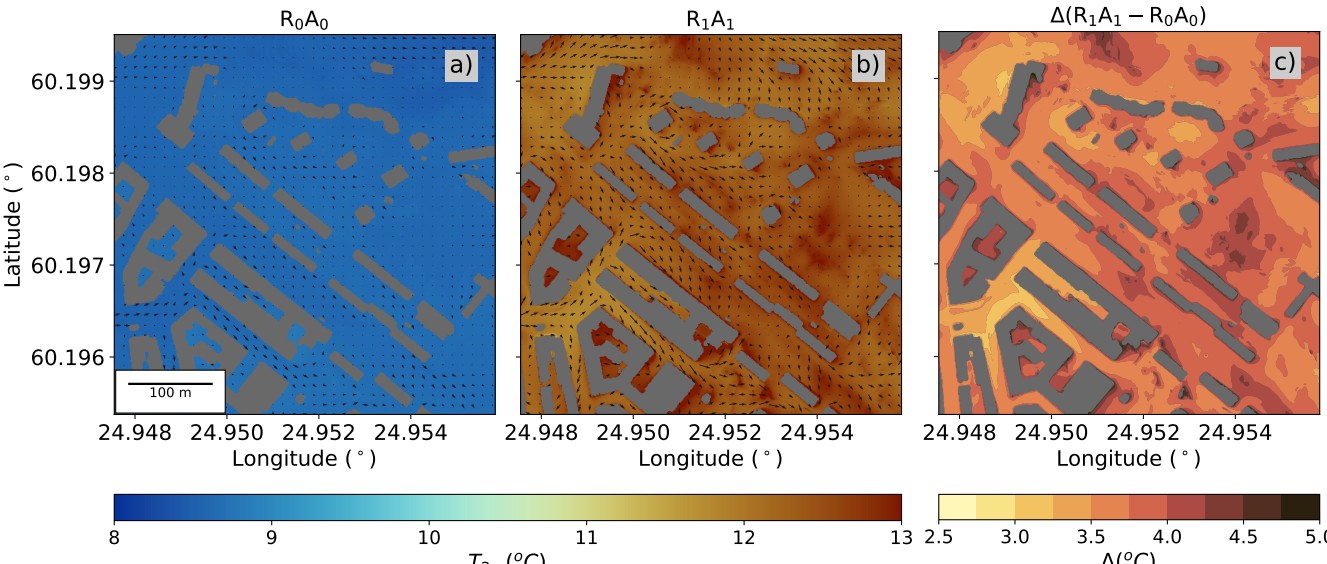

**Figure 3.** Near surface ($z = 2$ m) mean air temperature ($T_{2m}$, $^\circ$C) and flow field (arrows) for the base run $R_0A_0$ without radiation interaction and aerosol processes (a) and for run $R_1A_1$ with both radiation interaction and aerosol processes on (b), and the difference of $R_0A_0$ and $R_1A_1$ (c) averaged over 7:00-9:15 in the child domain.

(r4734) is known to overestimate heat input at vertical walls by roughly 20%, which can at least partially explain this near $1^\circ$C maximum difference (PALM Model System, 2021).

## 3.2 Flow field

In order to examine differences in the flow fields, $R_0A_0$ and $R_1A_1$ are examined at a height of 4-m ($V_{4m}$) (Figure 5). In the base run $R_0A_0$, the highest $V_{4m}$ are visible at the side street Southwest of the main street canyon reaching 1.3 m s$^{-1}$ and over an open flat terrain in the northern part of the child domain reaching 0.9 m s$^{-1}$ (Figure 5a). In $R_1A_1$, the flow field stays similar to $R_0A_0$ (Figure 5b) but an increase in the overall $V_{4m}$ is seen (Figure 5c). In this case, the smaller street canyon southwest of the main street canyon experiences stronger winds reaching 2.1 m s$^{-1}$. Some spots such as the eastern side of the main street show slight decrease in $V_{4m}$ by 0.6 m s$^{-1}$. The smallest difference in $V_{4m}$ between the two model runs is found at the location of the trees, where the tree canopies slow down the flow. Overall the mean flow increases by 89% in the child domain from 0.29 ($R_0A_0$) to 0.54 m s$^{-1}$ ($R_1A_1$) at the 4-m height due to enhanced circulation from radiative warming and cooling (Figure 6). Aerosol processes have minor impact on the flow causing a minor increase of 0.1% from $R_0A_0$ to $R_0A_1$ and 4.8% from $R_1A_0$ to $R_1A_1$ in $V_{4m}$. During the main run, the mean wind direction in the beginning of the simulation period is from the west but turns northwesterly during the simulation. The wind turning is greater in the base run compared to $R_1A_1$. The increase in wind speeds with heated surfaces has been commonly reported in previous studies. Li et al. (2010) used a ground heating approach and reported an increase in near ground flow and roof level streamwise flow with increasing instability. Vertical wind speeds

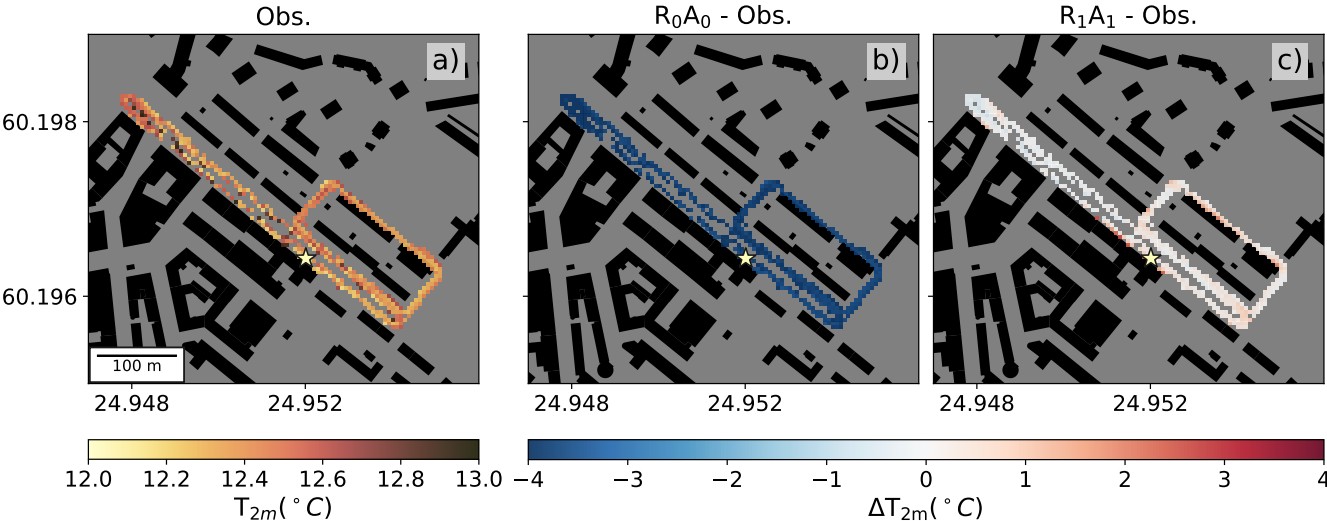

**Figure 4.** Observed 2-m air temperature ($T_{2m}$, °C) (a), the temperature difference between the base run ($R_0A_0$) without radiation interaction and aerosol processes on and observations (b), and the temperature difference between $R_1A_1$ with radiation interaction and aerosol processes on and observations (c). Negative values indicate underestimation and positive values overestimation of the observations. Supersite is marked with a yellow star.

showed an increase of up to 150%. Cheng and Liu (2011) reports a similar increase in mean flow speed at opposing sides of the
canyon of 100%, but additionally shows that the locations of the flow velocity maxima remain the same between neutral and
unstable cases. Similar observations about the locations of the flow maxima can be seen in Figure 6. Li et al. (2012) observed
a strengthening of the vortex due to buoyant lifting of leeward flow, which enhanced the rotation of the vortex and resulted in
150% increase in the windward vertical wind speeds. Nazarian et al. (2018) had similar wind speeds of 3 m s$^{-1}$ and reported
the street vortex becoming stronger and its centre moving towards the windward side. In these studies, no changes in wind
direction were seen likely due to fixed wind direction relative to idealised street canyons. In our simulations, the wind direction
changes during the simulation period presenting more realistic wind pattern.

Figure 6 shows the street canyon vortex within the main street canyon (see Figure 1) for $R_0A_0$ and $R_1A_1$. $R_1A_1$ shows the
effect of radiative forcing with stronger opposing wind speeds. Maximum ascent (descent) increased from 0.15 m s$^{-1}$ (-0.13 m
s$^{-1}$) in $R_0A_0$ to 0.69 m s$^{-1}$ (-0.33 m s$^{-1}$) in $R_1A_1$, due to radiative cooling and warming on opposite sides of the street canyon.
In the middle of the canyon, the effect of street trees is visible with enhanced ascent due to warming canopy, which spreads the
area of ascent more toward the middle of the canyon. Similar changes to vortex were reported by Xie et al. (2005) and Bottillo
et al. (2014) in idealised street canyon setups. As mentioned already above, the mean wind direction remains more westerly in
$R_1A_1$ when compared to $R_0A_0$. This increases the cross-flow component over the canyon in $R_1A_1$ and is one possible cause
for the more organized canyon vortex (Offerle et al., 2007; Dimitrova et al., 2009). This change in both the vortex structure
and strength in an unstable case compared to a neutral one has been shown in numerous studies (Nezis et al., 2011; Mei et al.,

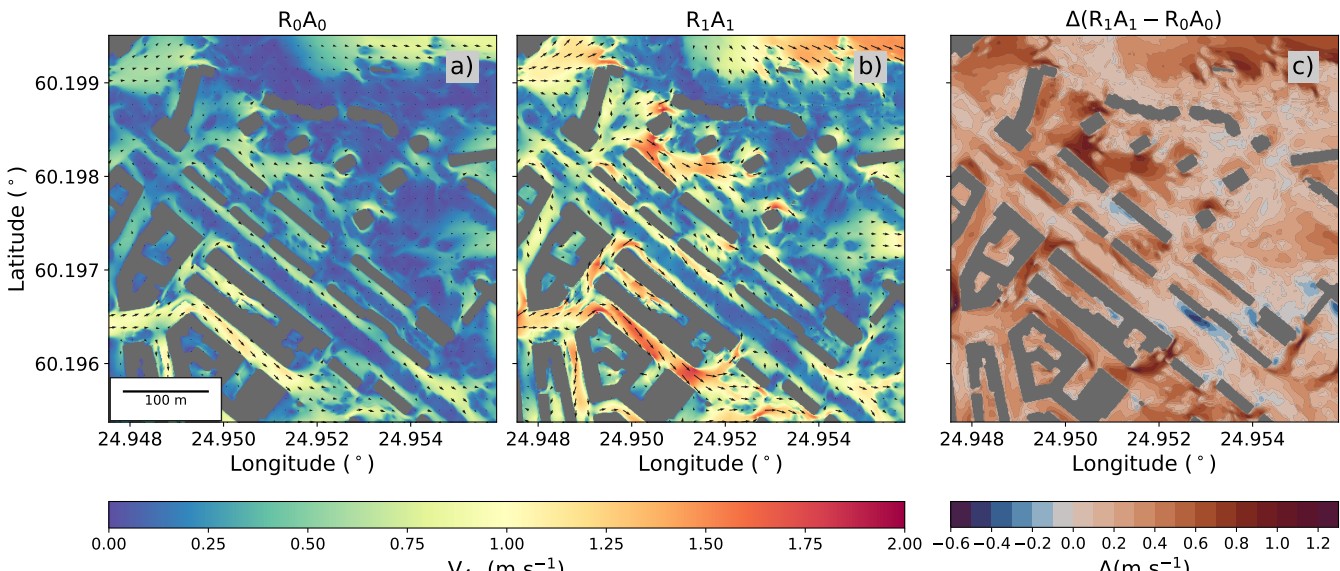

**Figure 5.** Mean horizontal wind speed at 4-m height ($V_{4m}$) between 07:00 - 09:15 for the base run ($R_0A_0$) when radiation effects are turned off (a) and for run with radiation effects turned on $R_1A_1$ (b), and the absolute change in horizontal wind speed between the two model runs (c).

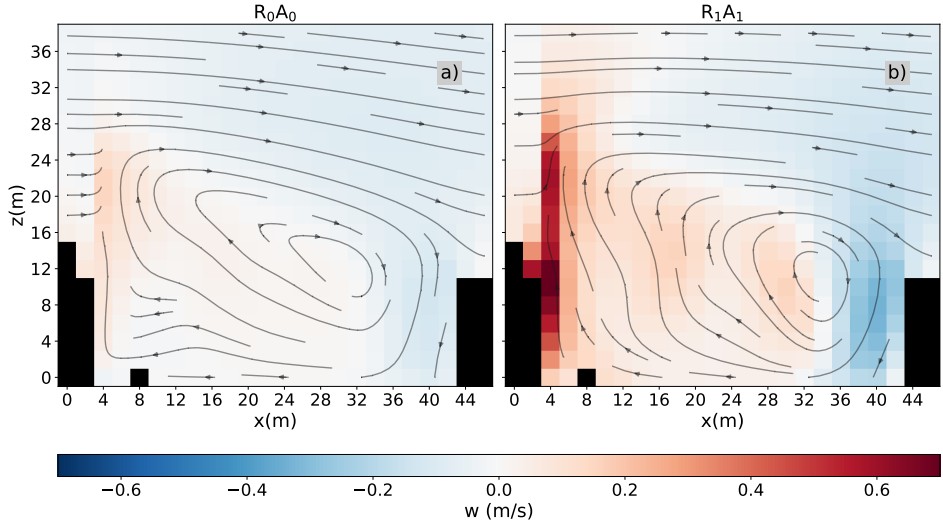

**Figure 6.** Street canyon vortex mean flow for the base run $R_0A_0$ with no radiation interaction (a) and $R_1A_1$ when the radiation model is turned on (b), averaged along the canyon over the shaded area shown in Figure 1. The streamlines describe the flow rotating around the y-axis aligned with the street and the colors describe the vertical wind speed. Black areas represent buildings on both sides of the street, with an additional measurement container on the western side.

2016; Guo et al., 2020). Previous studies suggest also that canyon vortices in wide street canyons are sensitive to the time of day, since heating of the windward wall would hinder the vortex instead of strengthening it, causing an entirely different flow structure in the canopy layer (Sini et al., 1996; Xie et al., 2005). As this is a calm wind case (mean wind 0.54 m s$^{-1}$), the effect of solar heating induced thermal turbulence has a larger effect on the flow than what would be with higher wind speeds (Bottillo et al., 2014). On the other hand, we simulate an early morning when solar radiation is around 240 W m$^{-2}$ when compared to midday radiation levels reaching 920 W m$^{-2}$ when the radiative heating is going to be even stronger.

### 3.3 Aerosol particle number concentration

The spatial variability of $N_{tot}$ at 4-m height ($N_{tot,4m}$) for the base run R$_0$A$_0$, and the differences of model runs R$_0$A$_1$, R$_1$A$_0$ and R$_1$A$_1$ compared to the base run are shown in Figure 7. R$_0$A$_0$ shows the largest concentrations on the Western side of the main street canyon reaching up to 155.4·10$^3$ cm$^{-3}$. This is due to the canyon vortex transporting traffic emissions to leeward side of the street canyon, which was also reported in previous studies. Nezis et al. (2011) reported pollutant concentrations having a direct correlation with the flow field and stability within the street canyon. This includes the leeward transport of pollutants within the canyon. Jiang and Yoshie (2018) found the temperature and flow distribution in an unstable case to also cause leeward transport of pollutants from the leeward side and that pollutant are removed from the canyon mainly at the sides of the canyon. Chen et al. (2020) focuses mainly on the temperature differences between eastward and westward facing walls during solar heating. They reported a high dependency of the street canyon orientation and aspect ratio on the resulting temperature distribution, which directly affects the flow conditions and ventilation. Kurppa et al. (2020) focused on mainly neutral cases and found the pollutant concentrations to be overestimated within the canyon when there was no heating present. Slightly smaller concentrations are seen when moving towards the southeast compared to other parts of the street canyon (Figure 7a). The side streets and surrounding areas away from the main street traffic emissions show the smallest concentrations staying above 3.6·10$^3$ cm$^{-3}$. When only aerosol processes are turned on (R$_0$A$_1$), there is generally a decrease in $N_{tot,4m}$ where trees in the main street canyon are located (Figure 7b). This is due to dry deposition removing particles from the air (Buccolieri et al., 2011; Karttunen et al., 2020). With only radiation interaction turned on (R$_1$A$_0$), $N_{tot,4}$ decreases on average by 53% from 15.7·10$^3$ cm$^{-3}$ to 7.0·10$^3$ cm$^{-3}$ in the child domain as the increased wind speed enhances the particle transport from the 4-m height upward (Figure 7c). An exception is seen close to building walls along the main street canyon in the central area. A small area of stagnating horizontal flow is formed in the middle close to the supersite (SR1), which combined with the overall pollutant transport to the leeward side of the street results in the largest increase in $N_{tot}$. The upward transport of pollutants starts at 4-m height and particles are then swept away above the canopy by the free flow.

Aerosol processes (R$_0$A$_1$) alone decrease $N_{tot,4m}$ in the child domain by 18%, while thermal turbulence alone (R$_1$A$_0$) decreases the concentrations by 53%. When the combined effect from radiation and aerosol processes (R$_1$A$_1$) are considered, $N_{tot,4m}$ is decreased by 56%. Previous studies such as Li et al. (2012) reported a decrease of 65% at the center of the canyon with aspect ratio of 0.5. Li et al. (2015) attributes this type of decrease as both due to turbulent and mean flow removal during unstable cases, whereas in a neutral case the mean flow is the main process of removing pollutants from the street canyon.

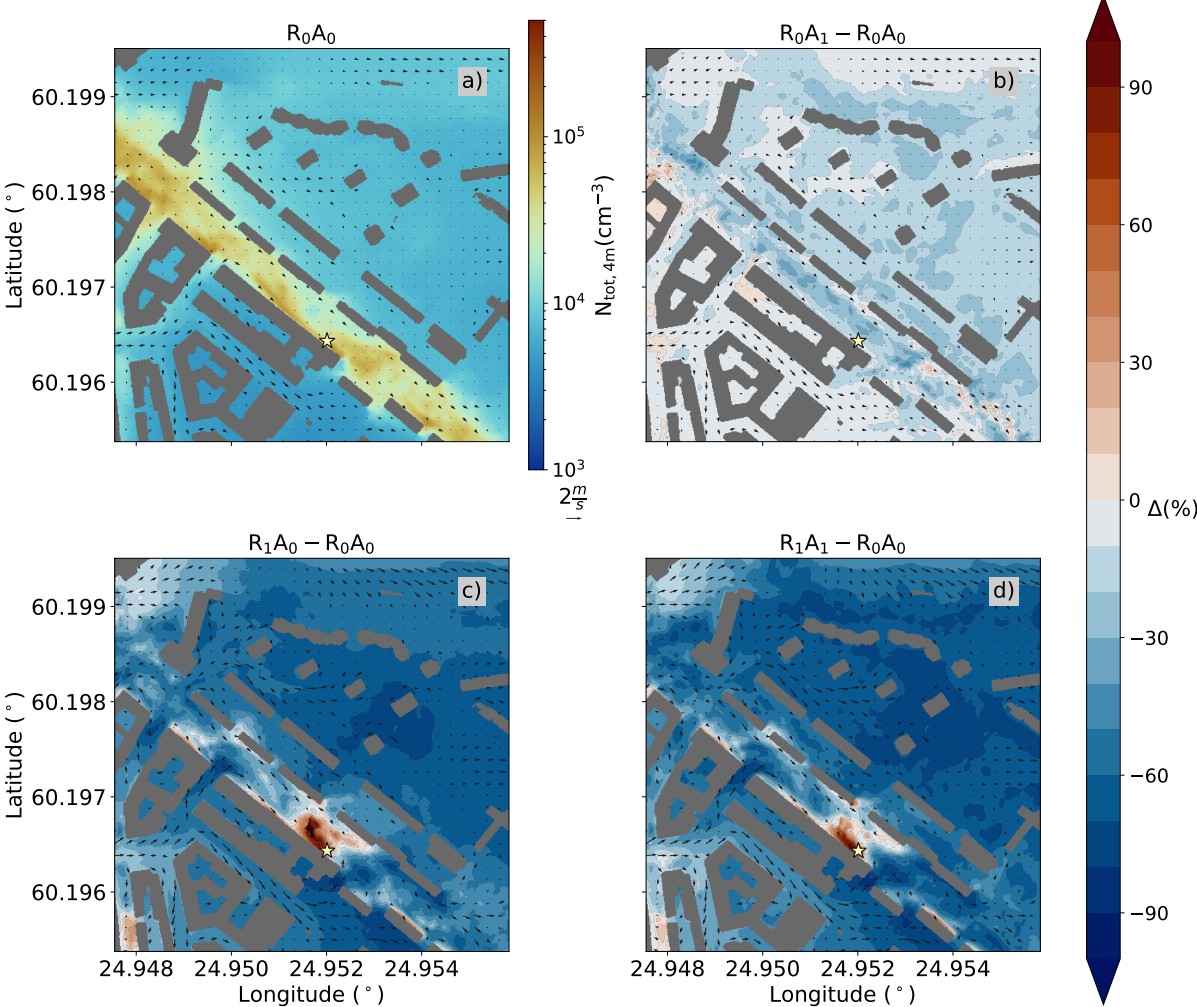

**Figure 7.** Mean total particle number concentration ($N_{tot,4m}$) and horizontal wind speed (arrows) at 4 metre height for the base run $R_0A_0$ (a), and the change in percentages of $R_0A_1$ (b), $R_1A_0$ (c) and $R_1A_1$ (d) compared to $R_0A_0$. Supersite is marked with a yellow star.

The modelled particle number concentrations at 2 metre height ($N_{tot,2m}$) are compared with the mobile laboratory measure-
ments in the child domain (Figure 8). The base run $R_0A_0$ and the run with only aerosol processes on ($R_0A_1$) show on average
98% and 74% larger $N_{tot,2m}$, respectively, than what is measured (Figure 8a). $R_0A_1$ performs better compared to $R_0A_0$ as the
effect of aerosol processes decreases the overall particle concentrations. When introducing radiation interaction, the model per-
forms much better, with 13% ($R_1A_0$) and 16% ($R_1A_1$) decrease in $N_{tot,2m}$ compared to observations. The enhanced pollutant
dispersion away from the surface decreases $N_{tot,2m}$ at nearly all locations, especially along the side streets. Thus, radiation
decreases also the absolute difference when compared to measurements and performs better compared to when radiation inter-
action is absent. The base run ($R_0A_0$) and the run with only aerosol processes on ($R_0A_1$) perform better in the southern sector

of the main street canyon, but due to the large overestimation at the northern end of the street, $R_1A_0$ and $R_1A_1$ perform better overall.

Figure 9 shows the averaged cross-section of $N_{tot}$ in the main street canyon. In the base run $R_0A_0$, the highest concentrations
reaching $104 \cdot 10^3$ cm$^{-3}$ are modelled at the ground level on the western side of the street canyon due to the street canyon vortex
(Figure 9a). Introducing the aerosol processes reduces $N_{tot}$ in the cross-section because of the combined effect of coagulation
scavenging the small particles, and dry deposition to the building and canopy surfaces (Figure 9b). The reduction is around
15.6% near the ground, and 21.0% within the street canyon (below 16 metres). When radiation is involved (Figure 9c), there is
a disparity between the amount of decreased pollutant concentrations when comparing the left and right side of the street, as the
transport across the canyon is more pronounced and the canyon vortex is modified. $R_1A_0$ shows an overall decrease of 27.1%
in the canyon $N_{tot}$ compared to the base run ($R_0A_0$), and near surface concentrations decrease by 40.5%. This infers that the
removal of pollutants is most effective near the ground at the centre of the canyon. Considering the combined effect of aerosol
processes and radiative heating in $R_1A_1$ (Figure 9d), a further decrease in particle concentrations appears in the middle and
especially eastern side of the main street canyon from ground level to the top of the tree canopy. The particle concentrations
are the lowest near the surface in the middle of the street with a decrease of 46.3%. Idealised simulations such as Xie et al.
(2005) reported stronger pollutant transport and vortex strength when the leeward canyon wall was heated, whereas ground
heating was more effective at pollutant removal overall. Nezis et al. (2011) shows similar results where the increased ascent
at the leeward side combined with the horizontal transport removes pollutants from the canyon and are transported away by
the flow at roof level. Mei et al. (2016) reported a similar one-vortex flow when the aspect ratio is 0.5, with direct correlation
between increasing instability and decrease in pollutant concentrations within the canyon. Mei et al. (2017) used a sinusoidal
function to model the thermal conditions in an idealised street canyon setup and found PM mass to decrease in the canyon with
increasing instability.

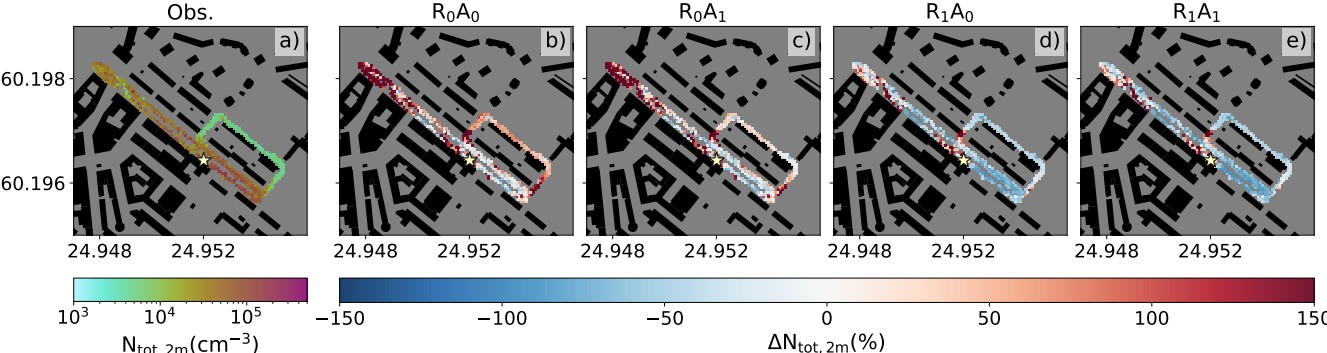

**Figure 8.** Observed 2m total particle number concentration ($N_{tot,2m}$) (a), and the difference of the base run ($R_0A_0$) (b), aerosol processes on ($R_0A_1$) (c), radiation interaction on ($R_1A_0$) (d) and both on ($R_1A_1$) (e) to the observed concentrations.

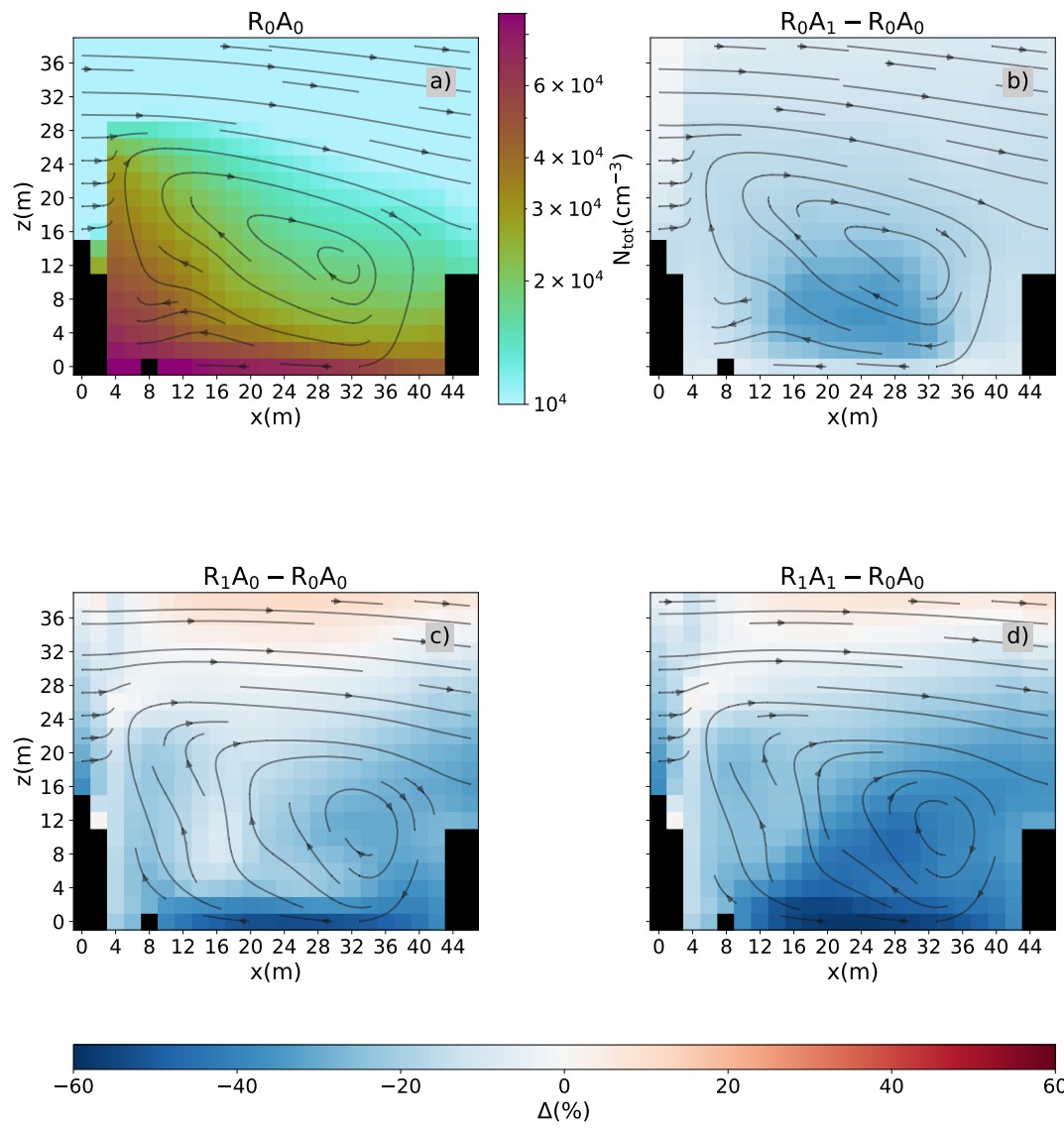

**Figure 9.** Mean total particle number concentration ($N_{tot}$) cross-section along the main street canyon in the base run $R_0A_0$ (a), and the difference of the model runs with only aerosol processes on ($R_0A_1$) (b), only radiation on ($R_1A_0$) (c) and both on ($R_1A_1$) (d) compared to the base run.

## 3.4 Aerosol particle size distribution

Focusing only on $N_{tot}$ ignores the effects of the aerosol dynamic processes on the different size of aerosol particles. Figure 10 shows the particle size distribution at the statistical regions (see Figure 1). The dominant particle size shifts to larger size when the aerosol processes are introduced. The largest overall change appears in the range of the smallest particles (4-15 nm) between $R_0A_0$ and $R_0A_1$. The number concentration of small particles is orders of magnitude larger when aerosol processes are off ($R_0A_0$ and $R_1A_0$) compared to that when aerosol processes are on ($R_0A_1$ and $R_1A_1$). This is due to coagulation and condensation which act as a sink for the smallest particles. A simulation in a narrow street canyon in Cambridge also indicated that aerosol processes have the greatest effect on the smallest particles independently of modelling height, and the condensation growth contributes much more to the reduction of small particle concentrations than coagulation (Kurppa et al., 2019).

At the supersite, the concentration of aerosol particles at all size bins is higher with radiation interaction on, especially for small particles, regardless of aerosol processes are introduced or not. The increase in small particle concentration can be up to 8.2 times in the case of $R_1A_1$ compared to $R_0A_1$. When radiation is introduced, the combined effect of the stronger transport towards the leeward wall and the stagnant flow parallel to the canyon lead to this increment in particle concentration. Similarly, the results of CFD demonstrated that the heating of the leeward façade further developed the clockwise-rotating vortex, and the pollutants would be brought to the leeward side, leading to a zone of higher pollutant concentration (Xie et al., 2007). At the opposite of the supersite, the effect of radiation on aerosol concentrations is not significant when the aerosol processes are absent. When the aerosol processes are switched on, the concentration of small particles is higher with radiation interaction on, which is probably related to the interaction of the flow field, temperature, and coagulation and condensation processes of small particles. Meanwhile, the background particle concentrations at all size bins are lower compared to the street canyon. Unlike the main street, particle concentrations in the background site are larger in $R_0A_0$ and $R_0A_1$. The inclusion of radiation interaction reduces particle concentration for all size bins. The background site is at the edge of a gravel football field, so street canyon flows such as an enhanced vortex structure and the increased ventilation caused by it are not as evident here compared to the main street. Xie et al. (2007) also pointed out that in a street canyon of H/W= 0.1, the wind structure was not isolated and involving heating did not lead to extreme pollution zones. In addition, the street canyon average particle size distribution (Figure 10d) resembles the distribution on the northeastern side of the street (SR2).

In our simulations, the particle deposition to vegetation is considered but not the possible impact of biogenic emissions to particle size distributions and concentrations. Primary biological aerosols (PBA) directly emitted from vegetation, such as spores and pollen, have commonly particle diameter larger than 1 $\mu$m (Fröhlich-Nowoisky et al., 2016). This exceeds the considered particle size range (2.5 nm - 1 $\mu$m) in our simulations and thus the lack of PBA is expected to have minor impact on simulated particle distributions. Biogenic volatile organic compounds (BVOCs) emitted from vegetation form secondary organic aerosols through gas-to-particle conversion (Schobesberger et al., 2013). The particle formation rate correlates positively with BVOCs (Dal Maso et al., 2016). Due to the lack of BVOC emissions, the current simulations may underestimate the concentration of small organic aerosols particularly in the southeastern part of the child domain where most vegetation is present. In addition, the lack of the condensation process of BVOCs would affect the particle size distribution in our results.

It has been shown that growth rates of small particles are correlated very well with total BVOC concentrations (Dal Maso et al., 2016). However, the measurement campaign made at the Helsinki supersite (SR1) shows that in this traffic environment, BVOC concentrations are significantly lower than anthropogenic VOCs (Saarikoski et al., 2023), and thus their lack of is likely to have relatively small impact to the simulated aerosol particle size distributions.

## 3.5 Vertical profiles of pollutant concentrations

The vertical aerosol profiles from both sides of the main street canyon (statistical regions SR1 and SR2) are illustrated in Figure 11 for the different aerosol metrics $N_{tot}$, UFP, PM$_{2.5}$ and LDSA. When radiation interaction is included in the simulations (R$_1$A$_0$ and R$_1$A$_1$), the removal of particles from the canyon is evident in all aerosol metrics at SR1. Particularly the metrics representing larger particles (LDSA and PM$_{2.5}$) decrease below the roof-top. A change in stratification from near neutral to unstable, and the resulting reduction in pollutant concentrations within street level concentrations has also been reported by previous studies. Nezis et al. (2011) focused mainly on structural flow field changes, whereas Mei et al. (2016) showed a decrease of 100% at the leeward side of the street for a 0.5 canyon aspect ratio. Jiang and Yoshie (2018) focused on inflow and outflow rates between the canyon and roof level. They observed low pollutant concentration air entering the canyon from the windward side and mixing with the polluted air, which combined with the lifting on the leeward side removes pollutants from the canyon. Above the roof-top the profiles of $N_{tot}$ and UFP are similar between the runs, whereas PM$_{2.5}$ and LDSA have more variability. Due to radiation interaction, PM$_{2.5}$ decrease on both R$_1$A$_0$ and R$_1$A$_1$ whereas LDSA decreases only in the absence of aerosol processes. In SR2, there is a slight reduction in street canyon concentrations of $N_{tot}$ and UFP, but a larger increase above the canopy in runs with radiation interaction on (R$_1$A$_0$ and R$_1$A$_1$). This suggests that the modified canyon circulation enhances the transport of smaller particles above the roof-tops on the windward of the street canyon. A similar behavior in concentrations increasing above the canopy due to the expanding canyon vortex was reported by Mei et al. (2016) in idealised simulations with unstable stratification and similar aspect ratio (0.5) compared to our case (0.45). PM$_{2.5}$ and LDSA have similar behaviour at SR2 than SR1 with PM$_{2.5}$ systematically decreasing, and LDSA decreasing with radiation interaction and increasing with aerosol processes. The transport towards the leeward side of the canyon is seen as higher concentrations at SR1 compared to SR2. Again the increase in the crossing flow component in R$_1$A$_0$ and R$_1$A$_1$ resulting in packing of the pollutants leeward side of the street canyon compared to R$_0$A$_0$ and R$_0$A$_1$ is visible.

Both at SR1 and SR2, the aerosol processes are more important (R$_0$A$_1$ and R$_1$A$_1$) for LDSA than for other aerosol metrics. Although the concentration of the smallest particles decreases, the increased concentration of particles larger than 20 nm due to aerosol processes is more important to LDSA (Kuula et al., 2020). Compared to the drone observations (Figure 11g,h), SR1 shows most agreement with R$_1$A$_0$ out of all the processes reducing LDSA at low levels, but is greater than any model runs above the canopy. This might indicate issues in the background forcing of the particles. At SR2 however, both R$_0$A$_1$ and R$_1$A$_1$ show better agreement with observations, which suggests that aerosol processes are more important on this side of the street canyon.

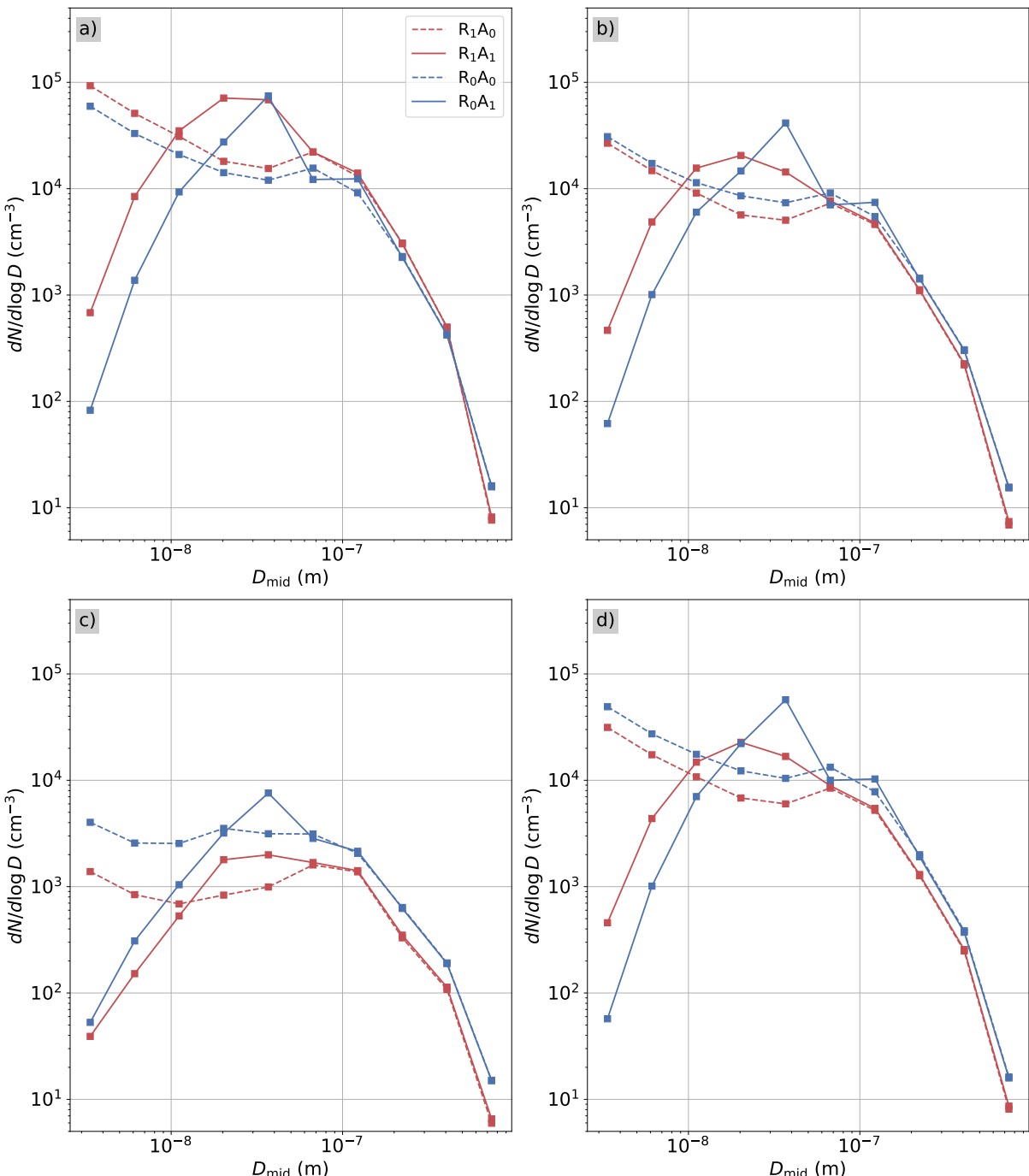

**Figure 10.** 4 metre mean particle number size distributions for the supersite (SR1) (a), opposite supersite (SR2) (b), background (SR3) (c) and canyon average (d) for runs with radiation ($R_1A_0$ and $R_1A_1$) and without radiation ($R_0A_0$ and $R_0A_1$).

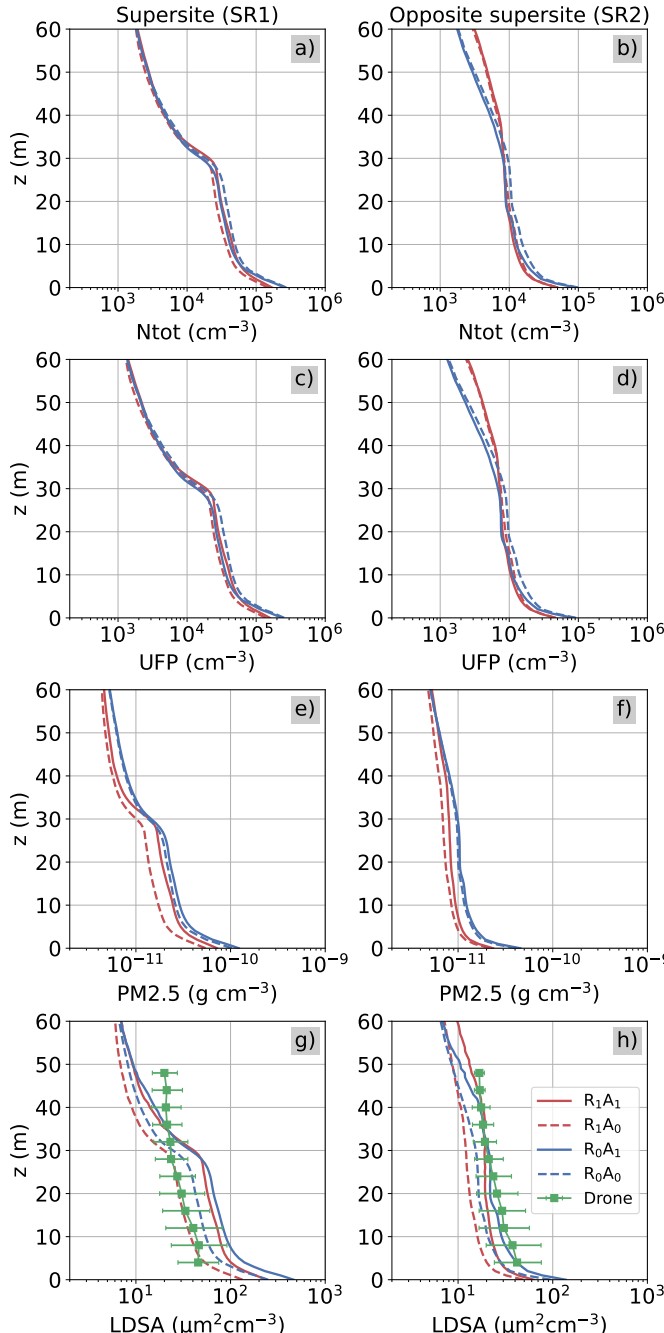

**Figure 11.** The geometric mean vertical profiles of total particle number concentration ($N_{tot}$) (a-b) number of ultrafine particles (UFP) (c-d), particle mass below 2.5 $\mu$m (PM$_{2.5}$) (e-f) and lung deposited surface area (LDSA) of particles (g-h) at the supersite (a,c,e,g) and opposite the supersite (b,d,f,h).

## 4 Conclusions

LES provides an optimal mean to examine flow and pollutant distributions in realistic urban areas as it can account for complex interactions between the surface and the air flow, radiation interaction and in some cases also aerosol particle dynamics, such as in the LES model PALM used in this study. The main aims of this study are to examine the impact of aerosol dynamics and radiation interaction on different aerosol metrics in real built-up neighbourhood in Helsinki. This was achieved using novel LES, which includes both aerosol processes and mixing conditions. The model performance was evaluated against near surface temperature ($T_{2m}$) and total aerosol particle number concentrations ($N_{tot}$) measured by a mobile laboratory, and lung deposited surface area (LDSA) measured by a drone. Four main runs to represent early summer morning on 6 June 2017 between 07:00 and 09:15 were performed. In the base run ($R_0A_0$) neither radiation interaction nor aerosol processes were on. $R_1A_0$ had only radiation interaction on, $R_0A_1$ only aerosol processes, and finally $R_1A_1$ radiation interaction and aerosol processes on.

In a calm wind case, such as the simulated summer morning, inclusion of radiation interaction improved the model performance in simulating the near-surface temperatures within the study area. In the base run, $T_{2m}$ was underestimated by on average 3.9°C. In $R_1A_1$, $T_{2m}$ was overestimated by 0.2°C, being on average 12.4°C. This change in temperatures and radiation provide energy for flow and the 4-m wind speeds increased on average from 0.29 m s$^{-1}$ to 0.55 m s$^{-1}$ within the study area.

Changes in flow increased ventilation and decreased particle concentrations close to the ground. The 4-meter $N_{tot}$ were reduced by 53% with radiation interaction included ($R_1A_0$). The inclusion of radiation interaction in LES is more important than adding the aerosol process which decreased the 4-m $N_{tot}$ concentrations by 18% ($R_0A_1$). Together with both the aerosol processes and radiation interaction included the concentrations decreased by 56 %. Compared to observations at the 2-meter modelling height, the near surface particle number concentration bias was reduced from 98% overestimation ($R_0A_0$) to 16% underestimation ($R_1A_1$). The bias is particularly reduced by inclusion of radiation interaction in the model runs.

Aerosol processes and their response to changes in flow altered the size distribution of particles. The size distribution in $R_0A_1$ and $R_1A_1$ showed larger particle sizes dominating, whereas in $R_0A_0$ and $R_1A_0$, the fraction of particles between 4-15 nm in diameter increased significantly due to the absence of processes such as deposition, coagulation and condensation. Radiation interaction and the enhanced flow field had a larger impact on the size distribution at the supersite, where the concentrations of all size bins increased by up to 8.2 times with $R_1A_1$ compared to $R_0A_1$ at the pedestrian level. Overall radiation interaction had the largest effect in medium to small particle size range at this height.

The change in stratification affected also the aerosol vertical profiles. All aerosol concentrations decrease in the street canyon when radiation interaction is considered, the effect being larger for PM$_{2.5}$ and LDSA on both sides of the canyon. Above the canopy $N_{tot}$ and UFP increase at the windward side of the canyon due to modified street canyon vortex by radiation interaction. Supersite shows higher concentrations overall compared to the opposite side due to the leeward transport described before. Aerosol processes have larger effect on the vertical profiles of PM$_{2.5}$ and LDSA than $N_{tot}$ and UFP, with the effect being particularly pronounced in LDSA. When taking into account both sides of the main canyon, $R_1A_0$ performs the best in terms of LDSA, as the change in flow alone is enough to bring LDSA closer to observations.

The results show that radiation interaction is more important to be considered in LES than aerosol processes when simulating pollutant distributions within urban neighbourhood in low wind conditions. Without radiation interaction, near surface air temperature and flow are underestimated and pollutant concentrations overestimated. Aerosol processes are however critical when aerosol particle size distributions, particularly the smallest size ranges, or vertical profiles of larger particles are examined. In our simulations with weak prevailing wind speed, the impact of radiation interaction on reducing the street-level concentrations can be greater than with stronger wind speeds. On the other hand, we simulated early morning when the radiative effects are not the strongest. In the future more meteorological conditions with varying wind speed and direction scenarios and time of day should be made to understand the effect of radiation interaction and aerosol processes in detail.

## Appendix A: Surface types used in the land surface model.

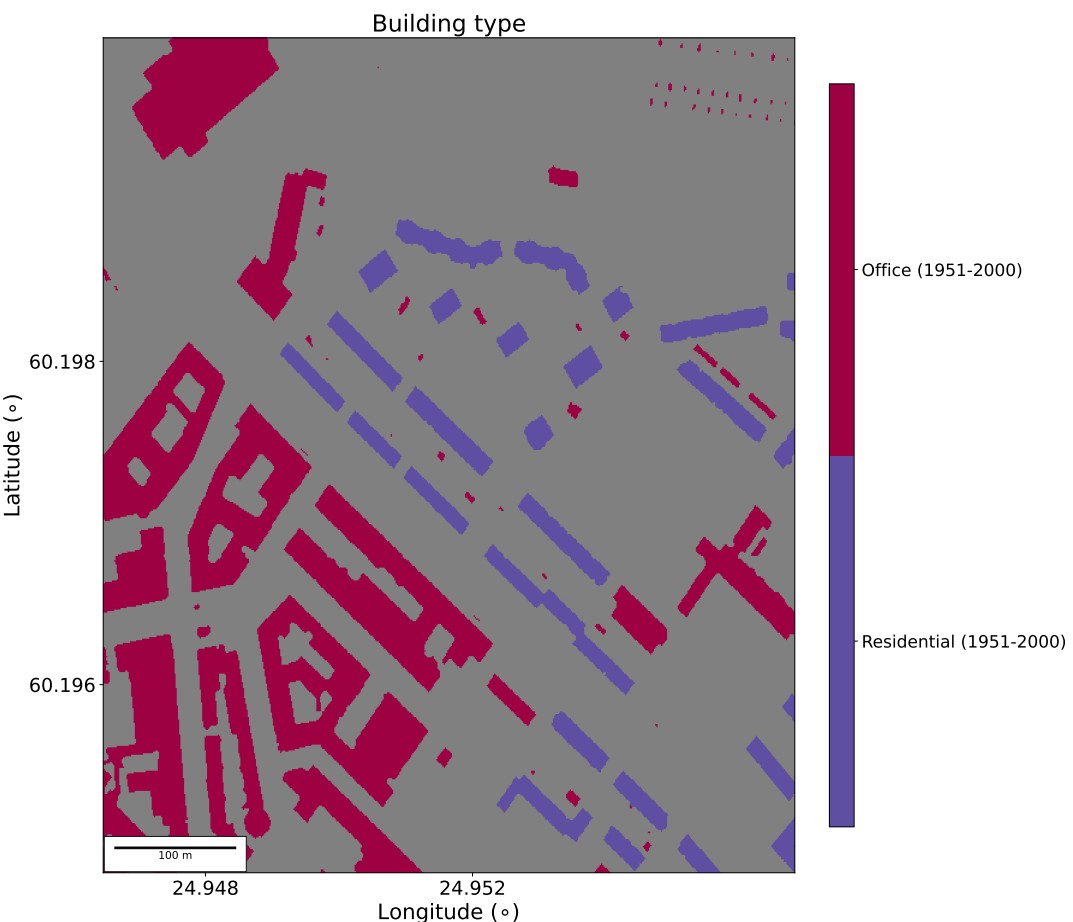

**Figure A1.** Building surface types in the child domain.

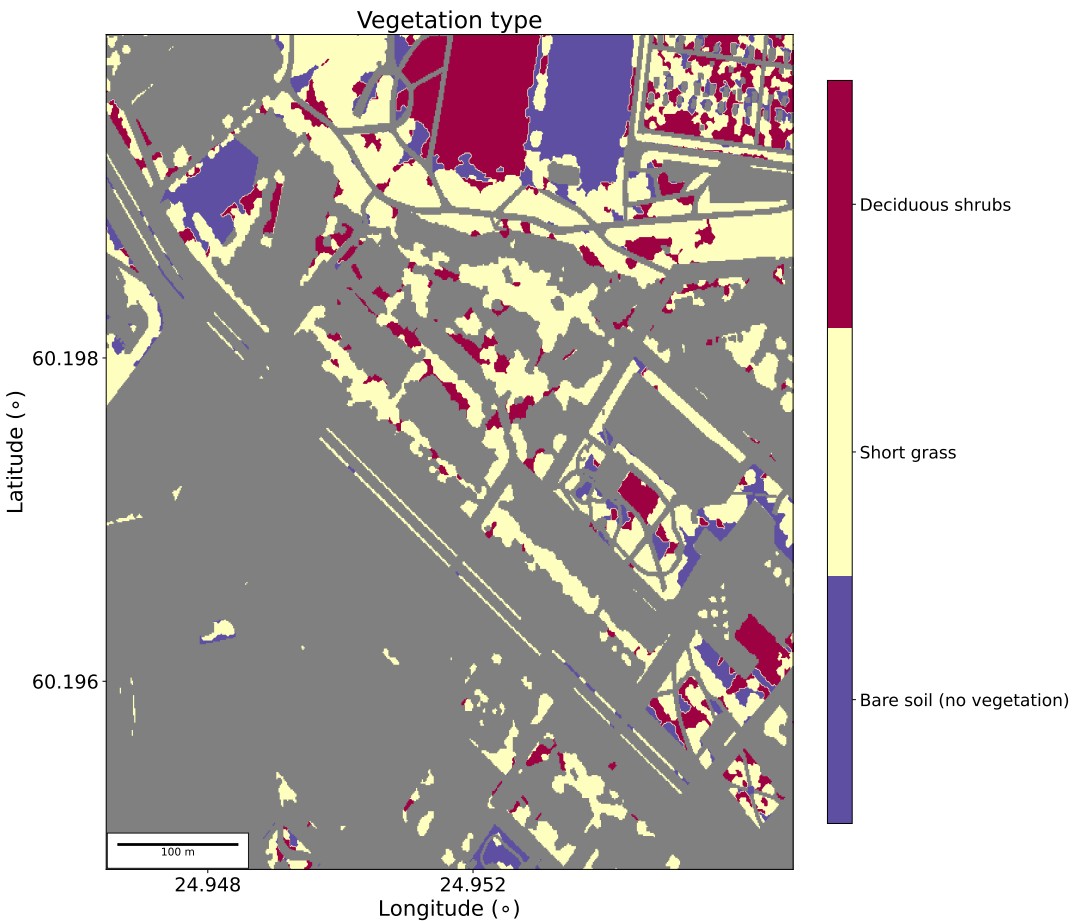

**Figure A2.** Vegetation surface types in the child domain.

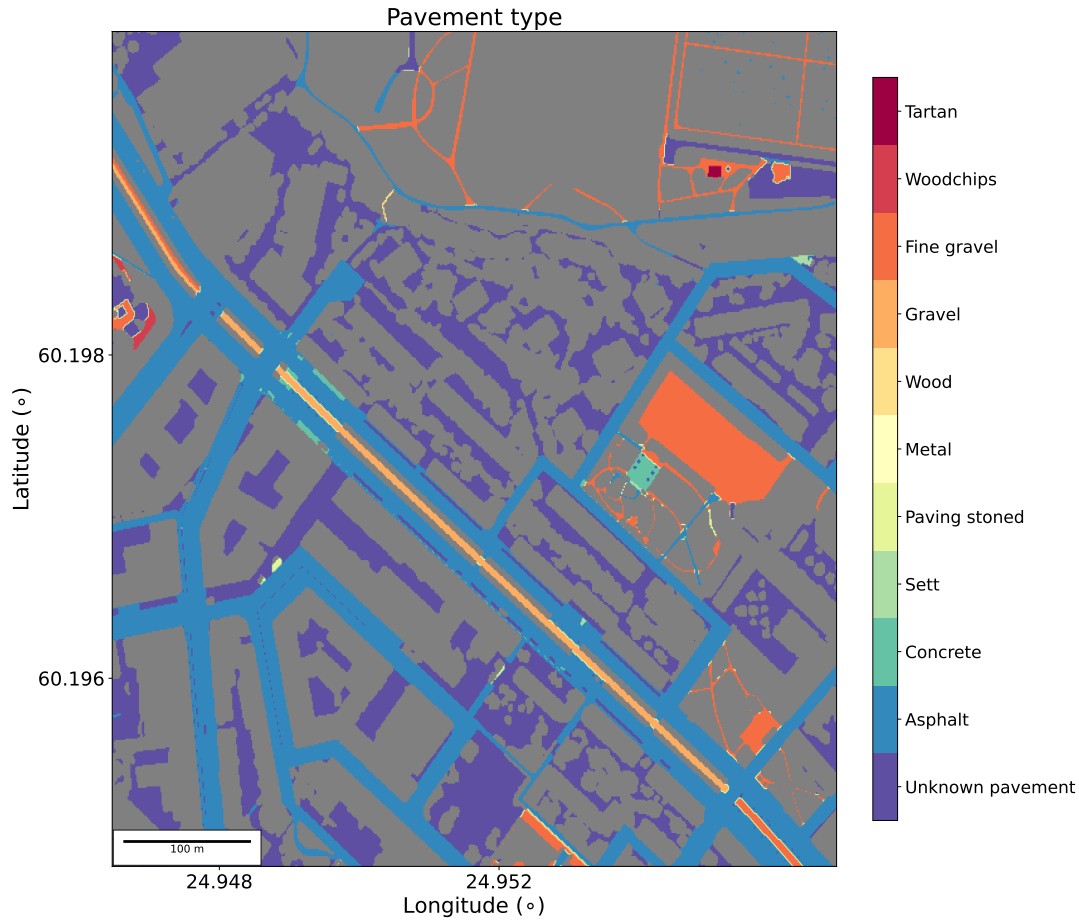

**Figure A3.** Pavement surface types in the child domain.

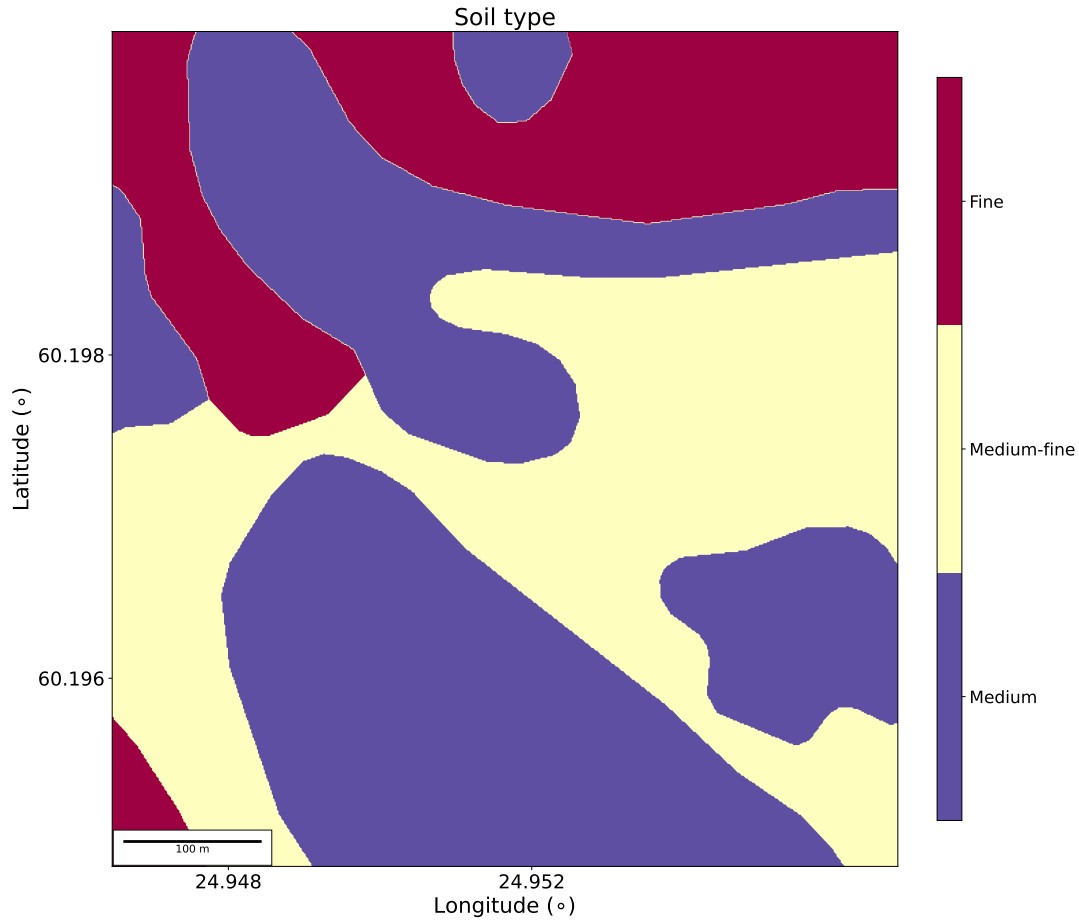

**Figure A4.** Soil surface types in the child domain.

*Code and data availability.* This manuscript uses input data and analysis scripts available from https://doi.org/10.5281/zenodo.7115705, https://doi.org/10.5281/zenodo.3839684 and https://doi.org/10.5281/zenodo.7124021

*Author contributions.* L.J., H.K. and L.P. planned and conducted the measurements. L.J., M.K. and J.S. planned the simulation setup. J.S. conducted the simulations. M.K. and J.S. wrote the code to analyse the data. J.S., L.J. and X.L. wrote the manuscript with input from all
425 authors. M.K., L.P. and H.K. commented on the manuscript. L.J. Supervised the project.

*Competing interests.* The authors declare that they have no conflict of interest.

*Acknowledgements.* We thank Helsinki Metropolitan Region Urban Research Program, the Academy of Finland (CousCOUS project, decision numbers: 332177 and 332178), the Academy of Finland ACCC Flagship (decision numbers: 337549, 337552 and 337551) and the Cityzer project funded by Tekes and Finnish companies (decision number: 2883/31/2015). This project has also received funding from the
430 European Union's Horizon 2020 research and innovation programme under grant agreement No 101036245 (project RI-URBANS) and 101037319 (project PAUL). The authors are very grateful to Aleksi Malinen and Sami Kulovuori from the Metropolia University of Applied Sciences for operation of the mobile laboratory Sniffer, and Aeromon Oy for conducting the drone measurements.

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
