# Peer review of "Effect of radiation interaction and aerosol processes on ventilation and aerosol concentrations in a real urban neighbourhood in Helsinki"

_EGUsphere, 2022_

## Author Comment (AC1)

We thank the reviewers for their valuable comments on the manuscript. Below are listed the changes made with comments from the authors. Line numbers correspond with the numbers in the revised paper. The reviewers' comments are bolded and our responses with normal text

**Reviewer #1**

**General**

**The paper examines the impact of thermal conditions and aerosol processes on local air pollutant concentrations in an urban environment using large eddy simulation (LES) model. It shows that including radiative interaction in LES improves the simulation of near-surface temperatures and ventilation of air pollutants, reducing the pedestrian level total particle number concentration. The inclusion of aerosol processes has a smaller effect. The study concludes that including radiative interaction and aerosol processes in LES is important for realistic simulation of near-surface aerosol particle concentrations.**

**The paper does not present a clear and sufficient level of novelty in the proposed approach and the model description (radiation part) is lacking in technical detail and clarity (see major issues).**

**Major issues**

**Novelty. The effect of solar radiation and surface thermal emissions of flow dynamics and pollutant dispersion is a topic that has been well studied in the field of urban climatology and wind engineering. In PALM's related publications in this topic the radiation effects are even discussed in more details where the individual components of radiative transfer processes are tested (Maronga et al., 2020; Krč et al., 2021; Salim et al., 2022). In this case findings of the current study lack novelty. Authors should cleatly show what is new in this paper and how it adds to the existing body of knowledge on the topic and how it may push the boundaries of the field in any substantial way.**

Response:
We acknowledge that the thermal effects and their impact on pollutant dispersion have been studied in the past also using PALM but these studies have treated air quality compounds as passive scalars without aerosol processes, and thus they represent simplified simulations. This is to our knowledge the first paper where thermal effects are jointly examined with aerosol processes within a real urban area. We have now emphasized this novelty in the abstract introduction and conclusions.

**Objectives.**

**Authors did not clearly state their objectives. Without clear objectives, it is difficult to understand the purpose of the study and how the research questions align with the overall goal of the paper. This lack of clarity can make it challenging for readers to understand the significance of the findings and how they contribute to the field of research. Furthermore, it may also make it difficult to understand the rationale behind the study design, making it hard to evaluate the methods used and the validity of the results. For example, authos should clearly state why it is important to know the effect of switch on/off the radiation and/or aerosol processes. Is it because the simplicity of the code, the computation time, the data availability, etc. Also, you need to justify why did you consider neighborhood in Helsinki (it is even in the paper's title).**

Response: We have now rephrased the objectives at the end of introduction (L53-58) and matched them with the novelty aspects of the manuscript (see response above).

It has not been fully understood whether or not including radiation interaction or aerosol processes is worth the computational cost or how large of an effect these two processes have on simulations. We have emphasized this in the text (L57-58): *"This is made as simplification in LES can save computational resources and for this it is important to understand the relative importance of different processes."*

Helsinki was selected due to the availability of large amounts of surface data and the intensive measurement campaign which was conducted within the simulation domain and is ideal for evaluating the model.. To emphasize this, we added sentence (L61-62) *"Helsinki was chosen due to the intensive observational air quality campaign made within the study area allowing extensive model evaluation."* at the end of the introduction section.

**Flaws in model description.**

**In Sec. 2.1.1 authors described RRTMG as the radiation model in PALM and they stated that it is capable of calculating multiple reflections, diffuse radiation and absorbed radiation on different surfaces. This is actually not accurate. Based on the radiation related publications for PALM (e.g.: Maronga et al., 2020; Krč et al., 2021; Salim et al., 2022), RRTMG is 1D external radiation model which is used to provide the radiation at each column in the domain for flat terrains. In case of obstacle, as in this case, RTM is used to calculate the radiative interactions within the urban area (urban surfaces and resolved vegetation). RRTMG itself is not capable to calculate multiple reflections, diffuse radiation and absorbed radiation on different surfaces. Having that**

**said, it is not clear how the run R0A0 is formulated in terms of radiation settings. Is it pure neutral case? Was RTM only switched off or both (RRTMG/RTM)? How LSM and USM were working in this case? Why we see temperature distribution then in case R0A0 (Fig. 3.a)?**

Response: The referee is correct. We have now corrected the terminology in section 2.1.1 and added more information about the radiation scheme.

The model runs without the radiation interaction represent neutral atmosphere and RRTMG/RTM were switched off. To make this more clear, we added a sentence describing the stratification in the different simulations to (P6, L135-137): *"The neutral simulations do not have USM and LSM since they require a radiation scheme. This means that the temperatures are directly provided by MEPS dynamic input in the neutral cases."*

**Generalization.**

**The paper has a significant drawback in that it lacks discussion of the findings. Without proper discussion, it is difficult to understand the implications of the results and how they relate to previous research in the field. Additionally, the lack of generalizability of the findings is a concern, as it limits the applicability of the study to a large audience. This may make it more difficult for others in the field to build upon the research and could hinder the advancement of knowledge in the area. Overall, the paper would benefit from a more thorough discussion and a clearer explanation of the generalizability of the findings. Section 3 reads as result section only and it does not contain adequate discussion. Also the conclusion section reads as a summary of the study and the results. Authors need to convince the readers that the lessons learned from the study are applicable for other model domains.**

Response:

Added more discussion to section 3 to open the previous studies' findings and how they relate to our results.

P10,L232-239

*Li et al. (2010) used a ground heating approach and reported an increase in near ground flow and roof level streamwise flow with increasing instability. Vertical wind speeds showed an increase of up to 150%. Cheng and Liu (2011) reports a similar*

*increase in mean flow speed at opposing sides of the canyon of 100%, but additionally shows that the locations of the flow velocity maxima remain the same between neutral and unstable cases. Similar observations about the locations of the flow maxima can be seen in Figure 6. Li et al. (2012) observed a strengthening of the vortex due to buoyant lifting of leeward flow, which enhanced the rotation of the vortex and resulted in 150% increase in the windward vertical wind speeds. Nazarian et al. (2018) had similar wind speeds of 3 m s$^{-1}$ and reported the street vortex becoming stronger and its centre moving towards the windward side.*

P13,L261-268

*Nezis et al. (2011) reported pollutant concentrations having a direct correlation with the flow field and stability within the street canyon. This includes the leeward transport of pollutants within the canyon. Jiang and Yoshie (2018) found the temperature and flow distribution in an unstable case to also cause leeward transport of pollutants from the leeward side and that pollutants are removed from the canyon mainly at the sides of the canyon. Chen et al. (2020) focuses mainly on the temperature differences between eastward and westward facing walls during solar heating. They reported a high dependency of the street canyon orientation and aspect ratio on the resulting temperature distribution, which directly affects the flow conditions. Kurppa et al. (2020) focused on mainly neutral cases and found the pollutant concentrations to be overestimated within the canyon when there was no heating present.*

P15-17,L305-312

*Idealised simulations such as Xie et al. (2005) reported stronger pollutant transport and vortex strength when the leeward canyon wall was heated, whereas ground heating was more effective at pollutant removal overall. Nezis et al. (2011) shows similar results where the increased ascent at the leeward side combined with the horizontal transport removes pollutants from the canyon and are transported away by the flow at roof level. Mei et al. (2016) reported a similar one-vortex flow when the aspect ratio is 0.5, with direct correlation between increasing instability and decrease in pollutant concentrations within the canyon. Mei et al. (2017) used a sinusoidal function to model the thermal conditions in an idealised street canyon setup and found PM mass to decrease in the canyon with increasing instability.*

P17,L319-350

**The abstract is quite long and reads more like a summary of the results rather than providing a concise overview of the research and its key findings. The**

**purpose of the research, the methods used, and the significance of the results should be more clearly stated in the abstract. The abstract should also provide a clearer and more comprehensive picture of the research context, problem and key findings.**

Response: The abstract has been shortened and we have also improved its clarity including description on novelty of the study.

**The figures present data plotted on a geographical coordinate (latitude and longitude) rather than a Cartesian grid. This makes it difficult to compare the data between different figures and to accurately measure the distances and areas depicted, especially in microscale simulations (limited domain size). I beleive that PALM uses Cartesian grid so did you projected the data before plotting and why? At lease add a scale in meter.**

Response: The base simulation of this study is based on the simulations from Kurppa et. al. (2020). The figures emphasized the geographical location of the street canyon and were made to be comparable to the earlier study. However, we have now added a meter scalebar to figures 1, 3, 4, 5 and appendix figures for easier measurement of distances.

**The reference Salim et al. 2020 should be updated to the final paper published in GMD not in the GMD(Discussion).**

Response: Reference has been updated (P30,L554-556)

**Page 9 line 121: I assume that aerosol processes may affect RRTMG and hence radiation inputs**

Response: In the current version of PALM, aerosols do not impact radiation transfer. This information was provided in P9, L204-205.

**Reviewer #2**

**In the study titled, 'Effect of radiation interaction and aerosol processes on ventilation and aerosol concentrations in a real urban neighbourhood in Helsinki', the authors use a large eddy simulation (LES) model to examine the impact of radiation interactions and aerosol processes for a real urban neighborhood in Helsinki. They find that the inclusion of radiation interactions largely improves model performance, particularly for temperature and aerosol concentrations. The study is well-designed and easy to read. I have some concerns that should be addressed before the manuscript is considered for publication.**

**That inclusion of radiation interactions would impact aerosols and near-surface meteorology within the urban canyon is expected and has been studied widely in the urban climatology literature, including in the LES literature. I am unclear what is new in this study other than the study area. A broader discussion of existing research gaps would be helpful**

Response: The novelty of the current study is that for the first time we combine both radiation interaction and aerosol processes in the same simulations within a real urban neighborhood. Previous studies have intensively studied the thermal effects on pollutant ventilation sometimes in idealized settings and sometimes within real urban neighborhoods, but these studies have been made using passive scalars. Different sizes of particles behave differently with flow and thus also implementing aerosol dynamics is relevant for realistic pollutant description. This is now emphasized throughout the manuscript.

**The authors note that the model considers both deposition on surfaces and emissions from road traffic. Is any emission from vegetation, such as of biogenic aerosols, considered in this study? If not, might be helpful to discuss. How would their lack of inclusion (if so) impact the results?**

Response: In this study, we only considered exhausted traffic emissions and did not involve emissions from vegetation. For primary biological aerosols (PBA), which are directly released from vegetation, such as spores and pollen, their particle diameter is usually larger than 1 μm (Fröhlich-Nowoisky et al., 2016). This exceeds the particle size range (2.5 nm - 1 μm) studied in our simulation. Therefore, the lack of PBA may not affect our results. Biogenic volatile compounds (BVOCs) emitted from vegetation, can form secondary organic aerosols through gas-to-particle conversion (Schobesberger et al., 2013). The particle formation rate correlates positively with the amount of BVOCs (Dal Maso et al., 2016). In the southeastern part of the model

child domain, there is a large distribution of vegetation, which can be treated as emission sources of BVOCs. Since the aerosol processes in the model do not include nucleation, the nucleation-mode aerosol concentrations were provided directly to the SALSA module as input. Due to the lack of vegetation emission, the current simulation may underestimate the concentration of small organic aerosols. In addition, the lack of the condensation process of BVOCs would affect the particle size distribution in our results. It has been shown that growth rates of small particles are correlated very well with total BVOC concentration (Dal Maso et al., 2016). However, a measurement campaign at the Helsinki supersite (SR1) showed that in a traffic environment, BVOC concentrations are significantly lower than anthropogenic VOCs (Saarikoski et al., 2023). Thus, it is feasible to ignore BVOC to some extent in this study.

We added discussion on the possible effects on ignoring biogenic emissions to the manuscript at the end of section 3.4 L338-350.

**The authors refer to the PALM-USM paper regarding radiation interactions between surfaces and the consideration of shading. This seems to be the version of the model used in this study, but the description is unclear. More information about how shading from buildings and trees are resolved would be helpful since shading can have large impact on temperatures within the urban canyon.**

Response: We updated the model description on how RTM handles shading differently between buildings and plant canopies (P3, L83-89):
*"RRTMG is used as a single-column model in PALM, whereas a separate multi-reflection RTM (Radiation Transfer Model) is used within the urban canopy layer (Resler et al., 2017). RRTMG feeds the RTM, which used by the surface models USM and LSM, with the necessary components such as the time of day and coordinates to solve the energy balance over all surfaces (Resler et al., 2017, Salim et al., 2022; Gehrke et al., 2020). RTM is capable of calculating multiple reflections, diffuse radiation and absorbed radiation on different surfaces (Krˇc et al., 2021). RTM handles plant canopies as fully transparent in the longwave radiation range and therefore shading is only modelled for the shortwave range in these cases (Resler et al., 2017)"*

**It would be good to have more information about the uncertainties and processing of the measurements. What are the specifications of these sensors? Was any quality control done when estimating composite values? Also, I do not think the air temperature measurement would really be for a 5 m x 5 m grid since the footprint of measurement is far greater (though more contributions come from nearby).**

Response: As suggested, we added more information about the specifications of the sensors to P8, L178-181. Data quality was naturally checked and from the standing measurements the first 3 mins were removed from the analysis. This information was now added. The 5x5 m grid is mainly determined by Ntot where this resolution makes sense and we wanted to use the same grids for both variables. We added a sentence "*The grid size was determined based on the concentration measurements and width of the streets*" on P9, L187-188.

**Minor comments:**

**Last line of abstract: Language is a bit unclear. Maybe mention 'for the case study of a calm period' in the previous line.**

Response: Improved the clarity of the lines mentioned (P1, L16-17).

**Line 25: Should be proportion, not number, of global population**

Response: Wording has been changed according to the suggestion (P2, L21).

**Table 1: Provide a column saying 'Model runs' above the four configurations**

Response: Added a title for the four configurations (P6, Table 1).

**Figure 4: Maybe in the caption, mention what the star represents (as done for Figure 7).**

Response: Added a sentence explaining the star (P11, Figure 4).

References:

Dal Maso, M., Liao, L., Wildt, J., Kiendler-Scharr, A., Kleist, E., Tillmann, R., Sipilä, M., Hakala, J., Lehtipalo, K., Ehn, M., Kerminen, V.-M., Kulmala, M., Worsnop, D., and Mentel, T.: A chamber study of the influence of boreal BVOC emissions and sulfuric acid on nanoparticle formation rates at ambient concentrations, Atmos. Chem. Phys., 16, 1955–1970, https://doi.org/10.5194/acp-16-1955-2016, 2016.

Fröhlich-Nowoisky, J., Kampf, C. J., Weber, B., Huffman, J. A., Pöhlker, C., Andreae, M. O., Lang-Yona, N., Burrows, S. M., Gunthe, S. S., Elbert, W., Su, H., Hoor, P., Thines, E., Hoffmann, T., Després, V. R., and Pöschl, U.: Bioaerosols in the Earth system: Climate, health, and ecosystem interactions, Atmospheric Research, 182, 346–376, https://doi.org/10.1016/j.atmosres.2016.07.018, 2016.

Kurppa, M., Roldin, P., Strömberg, J., Balling, A., Karttunen, S., Kuuluvainen, H., Niemi, J. V., Pirjola, L., Rönkkö, T., Timonen, H., Hellsten, A., and Järvi, L. (2020). Sensitivity of spatial aerosol particle distributions to the boundary conditions in the PALM model system 6.0. Geoscientific Model Development, 13(11):5663–5685.

Saarikoski, S., Hellén, H., Praplan, A. P., Schallhart, S., Clusius, P., Niemi, J. V., Kousa, A., Tykkä, T., Kouznetsov, R., Aurela, M., Salo, L., Rönkkö, T., Barreira, L. M. F., Pirjola, L., and Timonen, H.: Characterization of volatile organic compounds and submicron organic aerosol in a traffic environment, Atmos. Chem. Phys., 23, 2963–2982, https://doi.org/10.5194/acp-23-2963-2023, 2023.

Schobesberger, S., Junninen, H., Bianchi, F., Lönn, G., Ehn, M., Lehtipalo, K., Dommen, J., Ehrhart, S., Ortega, I. K., Franchin, A., Nieminen, T., Riccobono, F., Hutterli, M., Duplissy, J., Almeida, J., Amorim, A., Breitenlechner, M., Downard, A. J., Dunne, E. M., Flagan, R. C., Kajos, M., Keskinen, H., Kirkby, J., Kupc, A., Kürten, A., Kurtén, T., Laaksonen, A., Mathot, S., Onnela, A., Praplan, A. P., Rondo, L., Santos, F. D., Schallhart, S., Schnitzhofer, R., Sipilä, M., Tomé, A., Tsagkogeorgas, G., Vehkamäki, H., Wimmer, D., Baltensperger, U., Carslaw, K. S., Curtius, J., Hansel, A., Petäjä, T., Kulmala, M., Donahue, N. M., and Worsnop, D. R.: Molecular understanding of atmospheric particle formation from sulfuric acid and large